# Evolution of satellite plasmids can prolong the maintenance of newly acquired accessory genes in bacteria

Xue Zhang [1], Daniel E. Deatherage[1], Hao Zheng [2], Stratton J. Georgoulis[1] & Jeffrey E. Barrick [1]*

Transmissible plasmids spread genes encoding antibiotic resistance and other traits to new bacterial species. Here we report that laboratory populations of *Escherichia coli* with a newly acquired IncQ plasmid often evolve 'satellite plasmids' with deletions of accessory genes and genes required for plasmid replication. Satellite plasmids are molecular parasites: their presence reduces the copy number of the full-length plasmid on which they rely for their continued replication. Cells with satellite plasmids gain an immediate fitness advantage from reducing burdensome expression of accessory genes. Yet, they maintain copies of these genes and the complete plasmid, which potentially enables them to benefit from and transmit the traits they encode in the future. Evolution of satellite plasmids is transient. Cells that entirely lose accessory gene function or plasmid mobility dominate in the long run. Satellite plasmids also evolve in *Snodgrassella alvi* colonizing the honey bee gut, suggesting that this mechanism may broadly contribute to the importance of IncQ plasmids as agents of bacterial gene transfer in nature.

[1] Department of Molecular Biosciences, Center for Systems and Synthetic Biology, The University of Texas at Austin, Austin, TX 78712, USA. [2] Department of Integrative Biology, The University of Texas at Austin, Austin, TX 78712, USA. *email: jbarrick@cm.utexas.edu

Horizontal gene transfer (HGT) is a dominant genetic mechanism for disseminating novel traits, including antibiotic resistance, among microbial species and populations[1,2]. Conjugative and broad-host-range (BHR) plasmids are especially important for the transfer of genes between phylogenetically distant bacterial species[3]. Unlike the case for HGT via natural transformation or phage transduction, plasmid transfer by conjugation is not restricted by the requirement that DNA recombination and repair pathways integrate new genes into a bacterial chromosome. BHR plasmids, in particular, encode genes that carry out some of their own replication functions, which makes them less dependent on host factors than other plasmids and, therefore, more likely to successfully replicate in many types of bacterial cells. BHR plasmids can be classified into several different incompatibility groups on the basis of their replication mechanisms. Many IncP, IncN, IncW, and IncQ plasmids confer a wide spectrum of antibiotic resistance and are the most concerning with respect to the spread of multidrug resistance[4,5].

Recently acquired plasmids generally impose a fitness burden on a new host cell, which may be due to the cost of expressing antibiotic resistance genes[6] or due to plasmid-encoded genes interfering with host processes[7–9]. Therefore, purifying selection can rapidly lead to the loss of genes encoded on a plasmid from a new host unless there is positive selection for traits they confer[10]. Continuous transmission of a plasmid to new cells within a population can counteract the purifying selection process to some extent, but there is also a fitness cost of increasing the conjugation rate on donor cells[11]. Compensatory mutations on the chromosome and/or plasmid can sometimes ameliorate the cost of a plasmid and favor its persistence[12–19]. However, with some notable exceptions[16,19], it usually takes a long time—at least several hundred cell generations—and constant selection for plasmid function for compensatory mutations to arise when these mechanisms have been directly observed in laboratory populations of microbes[20]. These long timescales and stringent conditions may be unrealistic in most natural environments.

Most experimental studies of BHR plasmid stability have examined IncP plasmids[15–17]. Fewer studies have focused on IncQ plasmids, though these plasmids are found frequently in clinical and environmental settings and are known to confer a wide range of antibiotic resistance activities[21–24]. IncQ plasmids contain replication and mobilization modules in the backbone[25]. The RSF1010, R1162, and R300B plasmids isolated independently from *Escherichia coli*, *Pseudomonas aeruginosa*, and *Salmonella enterica* serovar *Typhimurium*, respectively, are the best-characterized members of this group[4,26,27]. In each of these plasmids, the minimal replicon consists of *repA*, *repB*, and *repC* genes, which encode helicase, primase, and iteron-binding proteins, respectively, and an *oriV* region containing the origin of replication[28]. Bidirectional replication of IncQ plasmids starts from the *ssiA* and *ssiB* initiation sites in *oriV* and occurs via an unusual strand displacement mechanism. In this replication mode, only one strand of DNA is replicated at a time from a given plasmid template molecule, while the other strand is displaced and released as single-stranded DNA (ssDNA)[25,29].

All IncQ plasmids encode multiple mobilization genes, including *mobA*, *mobB*, *mobC*, *mobD*, *mobE*, and the transmission initiation site, *oriT*[25]. They are mobilizable but not self-transmissible because they rely on mating pilus formation by a conjugative plasmid (e.g., an IncP or IncN plasmid) or by the host cell for DNA transfer[30,31]. During mobilization, the relaxosome nicks the plasmid at the *oriT* site and drags the ssDNA to the mating channel. Owing to their replication and mobilization mechanisms, most IncQ plasmid molecules exist in cells as ssDNA. For example, >70% of plasmid DNA was found to be single stranded by electron microscopy in R1162 plasmid preparations[32]. This single-stranded nature makes the plasmid prone to recombination, and this propensity is thought to restrict the size of IncQ plasmids. Natural isolates are typically in the 5–14 kb range and not larger[33].

IncQ plasmids with nearly identical sequences have been isolated from organisms at distant geographic locations[4,26,27], and identical IncQ plasmids have been isolated from the same environment at times separated by >30 years[23]. Both observations suggest that IncQ plasmid is evolutionarily stable. Bioinformatics analyses show that the accessory genes carried by mobilizable plasmids are often integrated into their sequences by processes mediated by other mobile genetic elements, such as transposons inserting into the plasmid, but that they can also be acquired directly by recombination mediated by short homologies[21,22,34]. The fitness burden of a plasmid on a host cell typically increases with the number of antibiotic resistance genes it carries[35]. The mechanisms underlying how IncQ plasmids are able to acquire and maintain these and other accessory genes are still not completely understood.

In this study, we examine the evolution of laboratory populations of *E. coli* with a newly acquired RSF1010-derived IncQ plasmid. Satellite plasmids (SPs) with deletions of accessory and replication genes rapidly arise that reduce the burden of plasmid carriage through a novel evolutionary mechanism. We characterize this process by monitoring accessory gene function with flow cytometry, by using next-generation sequencing to examine plasmid evolution, and by measuring the relative fitness of cells representing different endpoints of plasmid evolution. We also observe SP formation in *Snodgrassella alvi* wkB2, a bacterium from the honey bee (*Apis mellifera*) gut microbiome, when it is cultured in the laboratory and when it colonizes live bees. Finally, we examine the organization of this plasmid and perform population genetic simulations to understand how differences in mutation rates and the fitness effects of new mutant plasmids on cells and their progeny influence plasmid evolution. Our results suggest that the transient formation of SPs can favor the preservation of accessory genes encoded on an IncQ plasmid as it colonizes and spreads to new bacterial hosts.

## Results

**Nonautonomous IncQ satellite plasmids evolve in *E. coli*.** We used an experimental evolution approach to directly observe bacterial adaptation following acquisition of a BHR IncQ plasmid. Specifically, we added the plasmid pQGS (Fig. 1a), which has an RSF1010-derived backbone from pMMB67EH[36], to *E. coli* K-12 strain BW25113[37]. pQGS has three genes inserted into the RSF1010 backbone at a site where accessory genes are found in natural IncQ plasmids[33]. The constitutively expressed *gfp* gene and the adjacent *lacI* gene are not required for survival of these cells under the experimental conditions. They act as proxies for burdensome accessory genes. In a natural IncQ plasmid, such genes might encode resistance to an antibiotic or other stressor that is not yet present in the environment but could be in the future. pQGS also encodes the *aadA* spectinomycin-resistance gene at this site. Because we added spectinomycin to continuously select for maintenance of its function during the evolution experiment, *aadA* is an essential gene. It enables cells that acquired the plasmid to survive immediately, in their current environment, and cells without it will die. In the evolution experiment, six populations of *E. coli* BW25113 carrying pQGS (designated B1–B6) were serially transferred through 1:2000 daily dilutions into fresh medium such that they underwent ~330 total generations (cell doublings) over 28 total days of growth in liquid media.

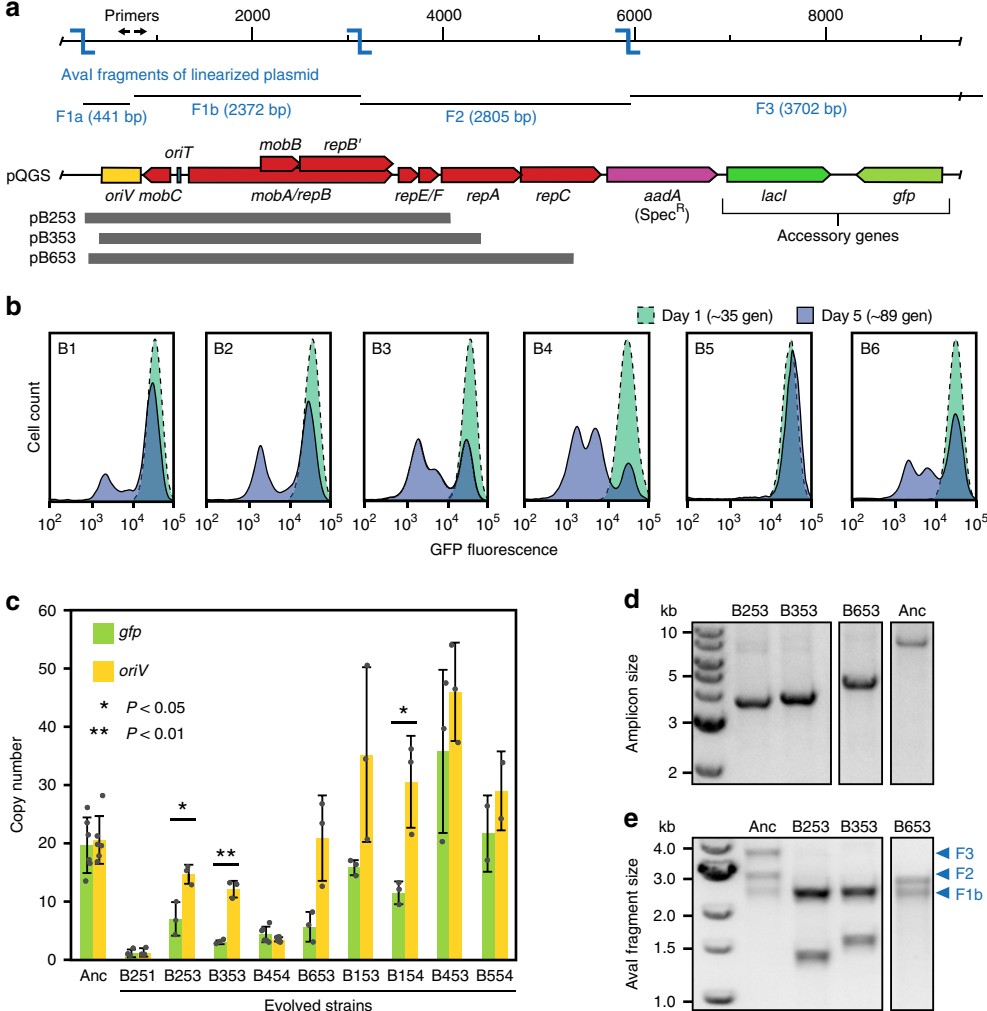

**Fig. 1 Satellite plasmids evolve from an IncQ plasmid in *E. coli*. a** Map of the pQGS plasmid used in evolution experiments. The *aadA* gene for Spec[R] was used to select for plasmid maintenance. Constitutively expressed *gfp* serves as a trackable proxy for a costly accessory gene. Primers used to generate PCR amplicons from pQGS for analysis, AvaI restriction sites (blue), and the fragments generated by digesting the linearized plasmid with AvaI (F1a, F2a, F2, and F3) are shown above the map. Plasmid regions preserved in SPs in evolved strains B253, B353, and B653 are pictured below. **b** Flow cytometry comparing the distributions of fluorescence in cell populations in all six experimental populations of *E. coli* after 1 and 5 days of evolution. **c** Copy number of the replication origin (*oriV*) and GFP gene (*gfp*) on the pQGS plasmid relative to a single-copy chromosomal reference gene (*dapA*) determined using quantitative PCR. The ancestor strain (Anc) and evolved strains from day 5 of the evolution experiment were tested. Error bars are standard deviations for three biological replicates except for ancestor and B454, which have six replicate measurements each, and B554, which has two. *P* values are for two-tailed *t* tests for rejecting the null hypothesis that *oriV* and *gfp* copy number are equal in a strain. **d** Linearized plasmid PCR amplicons from the ancestor strain (Anc) and three evolved strains that had significantly different *oriV* and *gfp* copy numbers. **e** DNA fragments produced by an AvaI restriction digest of the samples shown in **d**. The expected fragments are labeled in **a**. The smallest fragment (F1a) does not appear on the gel. For each SP fragment, F3 is missing and F2 is truncated to an extent that is consistent with identification of the precise boundaries of the deleted regions in each SP by Sanger sequencing. Source data are provided as a Source Data file.

We expected mutations would arise that eliminated expression of the *gfp* accessory gene to reduce the fitness burden of the plasmid on these new host cells. Therefore, we monitored green fluorescent protein (GFP) levels in the populations using flow cytometry. On day 1 of the evolution experiment, all six cell populations had a single uniform peak of highly fluorescent cells. By day 5, subpopulations of cells with reduced fluorescence had evolved and reached high frequencies in all but one population (Fig. 1b). However, we found no mutations in the *gfp* promoter or coding sequence in plasmids isolated from two random colonies with reduced fluorescence picked from each of the three different populations (B2, B3, and B6). Therefore, we next tested whether these strains had evolved decreased plasmid copy number, which could also reduce *gfp* expression and plasmid burden. Using

quantitative polymerase chain reaction (qPCR), we estimated the copy numbers of the plasmid origin of replication region (*oriV*) and *gfp* gene against a single-copy chromosomal reference gene (*dapA*) (Fig. 1c). Some clones had equally reduced *gfp* and *oriV* copy numbers. For example, strains B251 and B454 followed this pattern (evolved clones were designated with the letter B followed by the first digit indicating the population of origin, the second digit standing for the day of isolation, and the third identifying a specific clonal isolate). This result was consistent with the evolution of a reduced plasmid sequence copy number, but it did not entirely hold for all of the evolved strains. Others we examined (B253, B353, B154) had significantly different *gfp* and *oriV* copy numbers, always with more copies of *oriV* than *gfp* (Fig. 1c).

To figure out the reason for the discrepancy between *oriV* and *gfp* copy numbers in these strains, we further characterized plasmids isolated from three representative clones (B253, B353, B653). First, we checked plasmid size by electrophoresis. The IncQ plasmid is difficult to molecularly characterize due to its low copy number and the high prevalence of ssDNA resulting from its strand displacement replication mechanism[38]. Therefore, we used PCR with outward facing primers located inside *oriV* to create linearized amplicons for easier visualization (Fig. 1a). Compared to the ancestor, strains with different *oriV* and *gfp* copy numbers contained smaller plasmids (from 3 to 5 kb; Fig. 1d). The PCR amplicons were purified and digested with the AvaI restriction enzyme to determine roughly which regions of the plasmid were missing in these strains. For the ancestral plasmid, AvaI digestion of the linearized plasmid yields four fragments, one of which (F1a) is too small to visualize (Fig. 1a). Compared to the ancestral plasmid, all three of these evolved plasmids kept the *oriV*-bearing fragment (F1b), but the other two visible fragments (F2, F3) were either truncated or lost (Fig. 1e).

Sequencing the PCR amplicons allowed us to precisely localize the missing plasmid pieces. A region beginning within *repA* or *repC*, including the *gfp*, *lacI*, and *aadA* genes, and ending before *oriV* was deleted from each plasmid (Fig. 1a). The *repA* and *repC* genes encode the DNA helicase and iteron-binding protein, respectively, which are required for initiation of plasmid replication[28]. Loss of either function should lead to a plasmid that is unable to replicate on its own. As expected, attempts to transform the shorter plasmid variants on their own into new *E. coli* cells were unsuccessful. The PCR assay will preferentially amplify these smaller plasmids when they are present in a mixture, but weak bands corresponding to the linearized full-length pQGS plasmid are visible in the same sample in some cases (Fig. 1d). Since the qPCR results also show that copies of the *gfp* gene are still present in all three of these strains (Fig. 1c), we conclude that the smaller plasmids are satellite plasmids (SPs) that coexist with and require copies of the full-length plasmid in the same cell. We further confirmed that circular SP molecules form in these cells and are maintained alongside copies of the full-length ancestral plasmid by directly visualizing total plasmid DNA purified from B253, B353, and B653 cells by electrophoresis before and after linearizing the DNA molecules via cleavage at a single site with the ScaI restriction enzyme (Supplementary Fig. 1). From the perspective of the original plasmid, these SPs are parasites. They take advantage of its replication machinery without encoding all of the components needed for autonomous replication or the accessory genes.

We next examined how prevalent SP evolution was in our experiment. Using the PCR assay on DNA isolated from whole-population samples, we detected SPs in all 6 populations on day 5 (Supplementary Fig. 2). SP formation was so widespread that multiple different SPs were even observed in the same population in two cases (B1 and B4). Note, however, that owing to preferential amplification of smaller DNA fragments, the relative intensities of different bands in these PCR gels do not accurately reflect the representation of different molecular species in a sample. In particular, this comparison will overestimate the prevalence of SPs in a clone or population. Two to five clones from each population were isolated and the PCR assay was used to determine whether they harbored SPs found in the population results. Sequencing the PCR amplicons from isolates that contained one of these new SP variants revealed that they were similar to the initially characterized plasmids. All had deletions that completely eliminated the *gfp* and *aadA* genes, overlapped one or more *rep* genes, and left the *oriV* sequence intact (Supplementary Fig. 3). The deletions leading to SP formation were nearly always flanked by short, near-perfect sequence homologies with lengths of 7–15 base pairs (Supplementary Table 1), suggesting that SPs may form through RecA-independent processes[39].

**Satellite plasmids are transient evolutionary intermediates**. To understand the role of SPs in the maintenance of accessory genes, we further examined the time courses of plasmid evolution in the experimental populations. We found plasmids with deletions leading to 3.5–6.0 kb PCR amplicons had evolved in all 6 populations by day 5, persisted through day 9 in all but 1 population, and were still present on day 15 in 3 of the 6 populations (Fig. 2a, b and Supplementary Fig. 4a). Sequencing these PCR products showed that they were all nonautonomous SPs with deletions overlapping replication and accessory genes (Supplementary Fig. 2). In four populations (B3, B4, B5, B6), larger PCR amplicons with sizes of 7.0–8.0 kb emerged at later time points as SPs disappeared. These amplicons corresponded to various plasmids with accessory gene deletions that included loss of the entirety of *lacI* and a portion of *gfp* in every case (Supplementary Fig. 3). Complete sets of replication proteins and the *aadA* gene were retained in these plasmids, meaning that they remained fully autonomous. In the other two populations (B1 and B2), SPs disappeared, and no plasmid amplicon of any type was detectable on day 21 or day 25 (Fig. 2a and Supplementary Fig. 4a).

For two representative populations of each type (B2 and B6), we correlated flow cytometric data (Fig. 2c, d) with whole-genome sequencing (WGS) data (Fig. 2e, f) to characterize plasmid evolution in a way that was not subject to PCR amplification biases. Plasmid DNA isolated from cultures of the ancestral strain and whole-population samples archived at eight time points during the evolution experiment was examined by Illumina WGS. In addition to better defining evolutionary dynamics involving satellite and deletion plasmids (DPs), these data revealed an additional plasmid fate (Fig. 2g). In both populations, WGS reads consistent with an insertion of an IS*186* element into the pQGS plasmid *repF* gene were detected by day 6 (Fig. 2e, f). IS*186* is one of the most active insertion sequences (ISs) in *E. coli*. It inserted into pQGS at a site matching its preferred target site of 5′-GGGG(N$_6$/N$_7$)CCCC[40–42]. This IS insertion likely creates a second class of SP that is unable to replicate autonomously because it separates the *repA* and *repC* genes from their promoter, which is located upstream of the *repE* gene[28].

At the same time, we detected the IS*186* insertions, there was a precipitous drop in the copy number of the plasmid origin of replication (*oriV*) in both populations (Fig. 2e, f), and subpopulations of cells with greatly reduced fluorescence began to dominate (Fig. 2c, d). These results are consistent with integration of the IS*186*-disrupted plasmid sequences into the *E. coli* chromosome, either concomitant with IS*186* insertion or shortly thereafter, through single-crossover recombination with one of the three original copies of IS*186* (Fig. 2g). We were able to detect hybrid PCR products when using one chromosomal and one plasmid primer specific for one of the original IS*186* copies, indicating that this was the case. The presence of multiple peaks with intermediate fluorescence intensities—at the same time in population B6 (Fig. 2d) and later and to a lesser extent in population B2 (Fig. 2c)—indicates that the integrated plasmid sequence sometimes becomes multicopy within the chromosome. This could occur through homologous recombination during DNA replication, facilitated by the flanking IS*186* copies (Fig. 2g), leading to tandem duplications. Alternatively, plasmid sequences could integrate into multiple IS*186* copies or spread between them by gene conversion events. qPCR of the original isolates (Fig. 1c) found one or three equal copies of *gfp* and *oriV* in clones

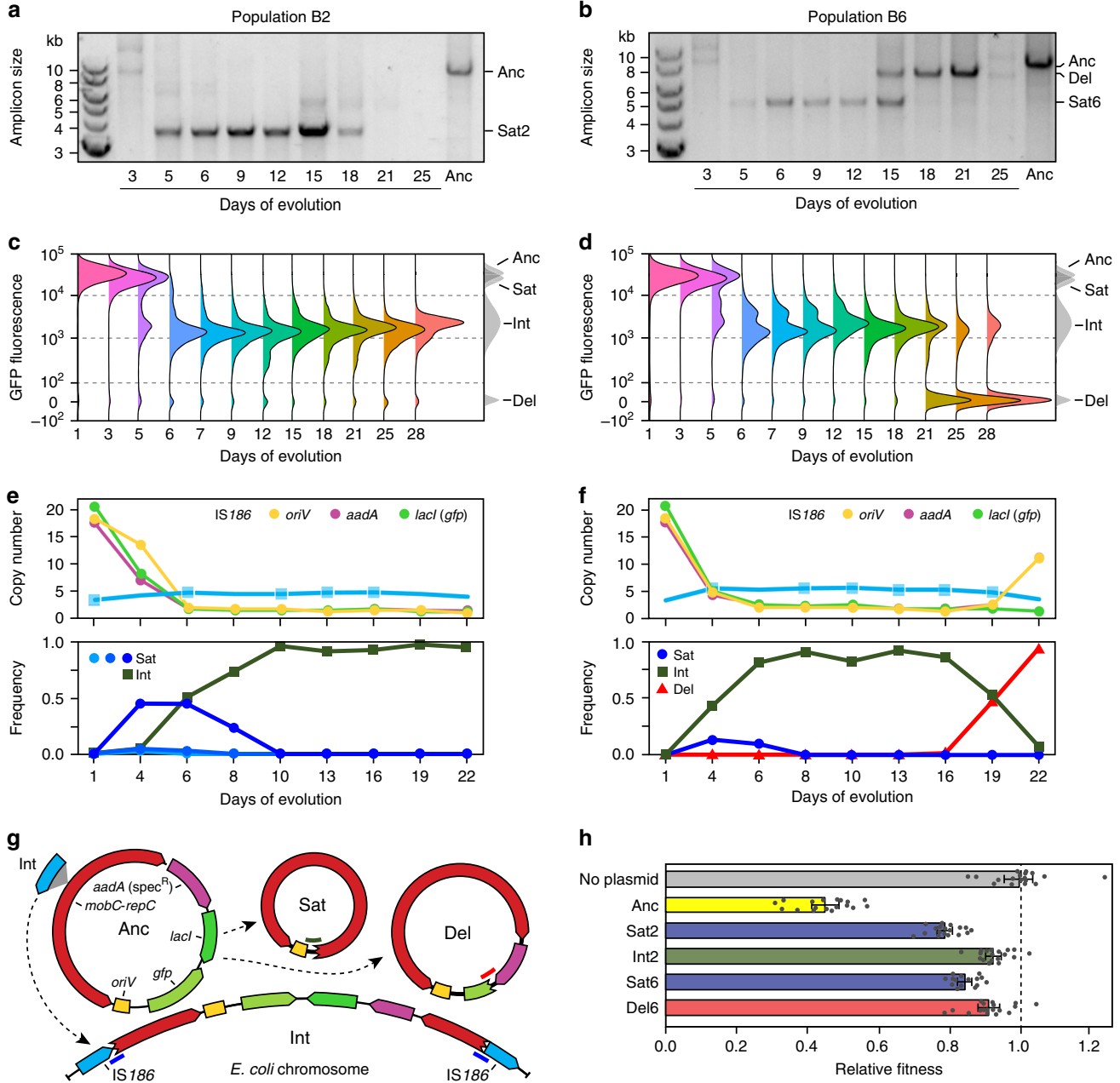

**Fig. 2 Time courses of plasmid evolution in *E. coli*. a, b** Appearance and persistence in populations B2 and B6 of satellite plasmids and other evolved plasmids with deletions that also change the size of the linearized plasmid PCR amplicon. Compared to the methods shown in other panels, the PCR assay is biased toward detecting satellite plasmids versus other plasmid fates even when they are very rare in a population because they produce smaller amplicons. **c, d** Distributions of GFP expression in cells in each population determined using flow cytometry. **e, f** Estimates of the copy numbers of different plasmid regions and the relative frequencies of different evolved plasmids' fates in each population from Illumina sequencing data. Coverage of the *lacI* gene was used to monitor the copy number of intact *gfp* gene copies because *lacI* is completely deleted in all evolved plasmids, but *gfp* is only partially deleted in some plasmids. The *x*-axes are categorical rather than linear in **c–f**, with even spacing of all samples that were analyzed from different days. **g** Schematic showing different evolved fates of the pQGS plasmid. Bold lines in blue, red, and dark green show the locations of sequencing reads unique to integrated, deletion, and satellite plasmids, respectively, which were used to estimate their relative frequencies. **h** Relative fitness of strains reconstructed with different evolved plasmid variants determined using co-culture competition assays versus a control strain with no plasmid. Error bars are 95% confidence intervals based on 15–18 replicate measurements for each strain. Source data are provided as a Source Data file.

B251 and B454 from day 5. Population B4 exhibits multiple and higher integrant fluorescence peaks at this time point (Supplementary Fig. 4b), whereas population B2 does not (Fig. 2c), which is consistent with this model.

We also tracked the prevalence of SPs from the numbers of WGS reads with split alignments that spanned deleted regions. SPs first appeared by day 4 in both populations (Fig. 2e, f). In

population B2, we could detect three different sizes of SPs. SP incidence as a percentage of all plasmids in the population peaked at ~50% in B2 and ~20% in B6. In population B2, the copy number estimated from the read-depth coverage of the accessory gene that is completely deleted in all SPs (*lacI*) dropped noticeably more quickly than that of the origin (*oriV*). This result is consistent with SPs and ancestral plasmids coexisting in a

sizable number of cells at the ratios observed by qPCR in clonal isolates from this population (Fig. 1c). SPs dropped below the detection limit by WGS at days 9 and 7 in populations B2 and B6, respectively. At this point, the IS-mediated integrants dominated. Completely nonfluorescent cells became dominant in population B6 near the end of the experiment (Fig. 2d). A new plasmid variant with a deletion of *lacI* and a portion of *gfp* was detected at day 19 by WGS in this population (Fig. 2f). According to the flow cytometric data, the prevalence of cells containing only the DP increased to 47% at day 21 and 66% at day 25.

Overall, these evolutionary trajectories suggest that SP evolution was beneficial to the fitness of a cell because it curtailed the expression of *gfp* and/or *lacI*. These results further suggest that IS186-mediated integration of the plasmid into the chromosome or deletion of these accessory genes from an evolved plasmid that remained capable of self-replication conferred greater fitness benefits than SP formation. Flow cytometric time courses show that these two fates eventually dominated in the other four populations of the evolution experiment (Supplementary Fig. 4b). However, owing to the long duration of the evolution experiment (~330 generations), other beneficial mutations are likely to have accumulated in the chromosomes of cells that have evolved plasmid sequences and influenced competition between each type by this time[43].

Therefore, to directly test the fitness consequences of evolving the different plasmid fates, we reconstructed cells of each type and competed them against a reference strain of *E. coli* BW25113 with the spectinomycin-resistance gene (*aadA*) integrated into its chromosome and a restored ability to utilize arabinose as a neutral genetic marker. First, we isolated SPs from each population (Sat2 and Sat6) and retransformed them into cells harboring the ancestral plasmid. Next, we isolated the DP from population B6 (Del6) and transformed it into the ancestral strain with no plasmid. Finally, we used transduction to move an integrated copy of the plasmid from an evolved cell in population B2 to the ancestral strain (Int2) so that its fitness could be measured without interference from any other evolved alleles.

As a control, we competed the plasmid-free ancestor strain BW25113 against the reference strain in media that did not contain any antibiotic. The fitness of the ancestral strain (No plasmid) was not significantly different from that of the reference strain ($p = 0.768$, two-tailed $t$ test). The ancestral strain/plasmid combination of BW25113/pQGS (Anc) had a greatly reduced fitness of 0.431 relative to the reference strain with no plasmid. The addition of a SP to a cell significantly alleviated this cost, with a relative fitness of 0.781 and 0.840 for Sat2 and Sat6, respectively (Fig. 2h). The cases of integration into the chromosome (Int2) or deletion of the accessory genes from the plasmid that we tested (Del6) restored fitness even more toward that of cells with no plasmid, to relative fitnesses of 0.923 and 0.910, respectively. As predicted by the coexistence of cells with DPs and cells with chromosomally integrated plasmids in population B6, the fitness values of these evolved plasmid fates were not significantly different ($p = 0.474$, two-tailed $t$ test). All strains with evolved plasmid variants were significantly more fit than the ancestor with pQGS ($p \leq 3.6 \times 10^{-14}$, Bonferroni-corrected two-tailed $t$ tests) and less fit than the reference strain with no plasmid ($p \leq 0.011$, Bonferroni-corrected two-tailed $t$ tests).

**Satellite plasmids evolve in a honey bee gut symbiont.** Given the wide host range of IncQ plasmids, we were interested in whether SPs would still evolve in a different bacterial species and in a more complex environment associated with an animal host. Honey bees (*A. mellifera*) have a simple and conserved gut microbiome[44], and we previously genetically engineered several

of its bacterial gut symbionts using RSF1010 plasmids[45]. We tested whether SP evolution occurred in *S. alvi* wkB2, a β-proteobacterium that has only recently been cultured[46]. We first checked for the evolution of SPs from pQGS within five populations of *S. alvi* wkB2 (designated S1–S5) that we serially passaged in vitro. We found mixtures of smaller amplicons indicative of plasmids with deletions when we used the PCR linearization assay on DNA isolated from these populations (Fig. 3a). By sequencing the PCR products, we determined that two identical bands detected at day 8 in all populations (pS621 and pS521) corresponded to SPs (Supplementary Fig. 3). Bands that likely correspond to smaller SPs (e.g., pS631) became dominant by day 12 in all populations. Larger amplicons that may be accessory gene DPs were also observed in two populations at this point.

Next, we examined whether SPs would evolve and persist in *S. alvi* wkB2 carrying pQGS when they colonized microbiota-free bees[47,48]. As expected from previous studies[49], *S. alvi* with the pQGS plasmid began to colonize the inner wall of the ileum by 24 h post-inoculation as visualized by GFP fluorescence in the bee gut (Fig. 3b). We further confirmed colonization by measuring colony-forming units (CFUs) in the guts isolated from sacrificed bees. On days 3 and 4, there were approximately $10^7$ and $10^8$ CFUs per bee gut, respectively. Previous studies have shown that *S. alvi* fully colonizes microbiota-free bees by 4–6 days post-inoculation[50]. We monitored SP formation in *S. alvi* carrying pQGS after it was inoculated into cohorts of bees housed in ten separate enclosures. Every 24 h for the first 4 days, one bee from each enclosure was sacrificed, and the PCR assay was performed on DNA isolated from its entire gut population. For 6/10 enclosures, we detected amplicons with sizes compatible with SPs in one or more of the bees that were sacrificed at these time points (Supplementary Fig. 5). We used Sanger sequencing to verify that three of these bands corresponded to SPs (pC562, pC765, pC1025) (Fig. 3c) and to precisely map their deletions (Supplementary Table 1). No SPs were detected by the PCR assay in additional bees that were sacrificed from these 10 enclosures after 8 and 12 days. These results indicate that SPs are a transient evolutionary intermediate in *S. alvi* colonizing the honey bee gut, as they were in *E. coli* and *S. alvi* passaged in vitro.

**Multiple factors favor satellite plasmid evolution.** Mutations that eliminate different regions of the pQGS plasmid can result in nonautonomous SPs or in autonomous plasmids that have deleted just the accessory genes. We found that *E. coli* cells with accessory gene DPs have a much higher fitness than cells containing a mixture of satellite and ancestor plasmids. If both types of cells arise at similar rates in a bacterial population, then cells with DPs are expected to dominate to such an extent that SPs should never reach an observable frequency, but this was not the case in our experiments. To investigate this discrepancy, we examined ways in which the genetic organization of the pQGS plasmid and its multicopy nature might favor SP evolution and thereby maintenance of the newly acquired accessory genes in a population.

First, there might be a mutational bias favoring the evolution of SPs. DPs must maintain the entire plasmid from *oriV* to *aadA* intact so that they remain autonomous and still encode antibiotic resistance. If we also assume that they must delete at least a portion of the both the *lacI*- and *gfp*-coding sequences to realize their full fitness advantage, then this constrains these deletions to relatively few combinations of starting and ending base positions. On the other hand, SPs must only maintain *oriV* while deleting the accessory genes plus *aadA* and at least one of the replication genes. Given just the raw numbers of start–end coordinate combinations fitting each of these scenarios (Fig. 4a), SPs would be expected to arise at 4.0 times the rate of DPs.

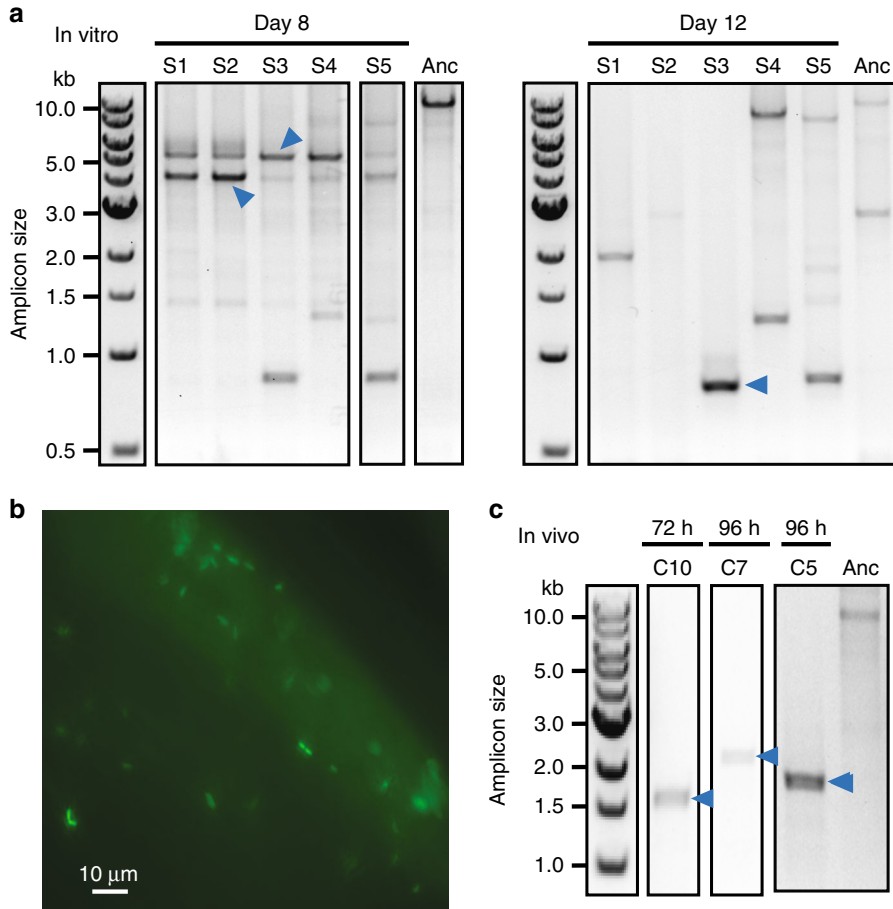

**Fig. 3 Satellite plasmids evolve in _S. alvi_ in the honey bee gut. a** SPs detected as linearized PCR amplicons during the in vitro propagation of _S. alvi_ at days 8 and 12 in five experimental populations S1–S5. **b** Fluorescence micrograph showing colonization of the bee gut ileum 24 h after inoculation with _S. alvi_ carrying plasmid pQGS. **c** Satellite plasmids detected as linearized PCR amplicons from DNA extracts of the guts of individual bees reared in different enclosures (C1–C10) at 72 or 96 h after inoculation. Full results for all populations evolved in vivo are shown in Supplementary Fig. 4. Blue arrows in all panels are bands that were Sanger sequenced to validate that they represent SPs (Supplementary Fig. 3 and Supplementary Table 1). Source data are provided as a Source Data file.

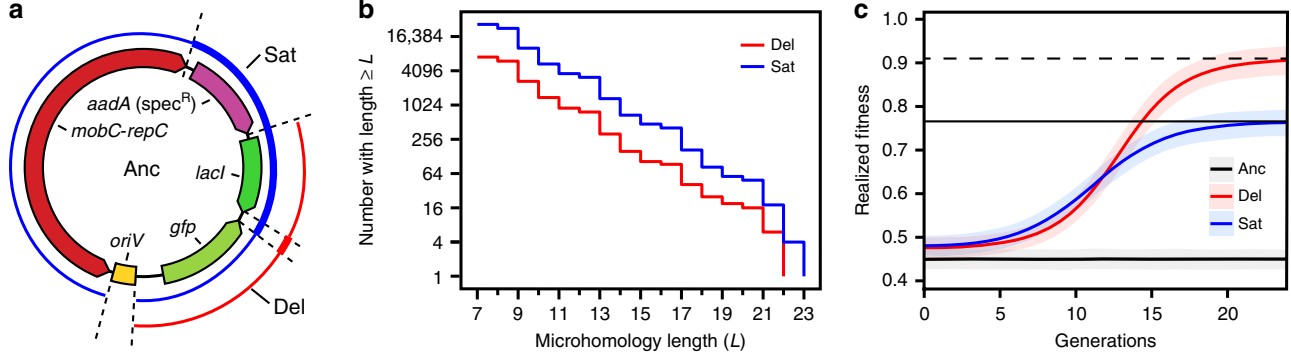

**Fig. 4 Satellite plasmid evolution is favored over deletion plasmid evolution. a** pQGS plasmid map showing regions containing the endpoints of deletions leading to satellite plasmids (blue thin line) and accessory gene deletion plasmids (red thin line). The regions required to be contained in each deletion are shown with bold lines in the respective colors. **b** Microhomology regions like those observed flanking satellite plasmid deletions are more common in the satellite plasmid deletion endpoint regions than in the accessory gene deletion plasmid endpoint regions, but the difference is no greater than the fourfold difference expected if these sites are randomly distributed. **c** Multilevel stochastic simulations predict an early advantage for cells with newly evolved satellite plasmids due to a reduced phenotypic lag. Their progeny realize more of their full fitness increase earlier, compared to cells that evolve deletion plasmids. Shaded areas are 95% confidence intervals from 200 simulation trajectories. Source data are provided as a Source Data file.

All 10 unique SPs we characterized resulted from deletions that had DNA sequence microhomologies consisting of near-exact 7–15 base pair repeats overlapping their endpoints in the ancestral plasmid (Supplementary Table 1). We investigated whether there was any bias in the abundance of similar microhomologies in different regions of the plasmid that might additionally favor deletions leading to SPs versus DPs (see "Methods"). Over a range of different microhomology lengths, the number of near-repeats that could mediate the formation of each type of evolved plasmid did not noticeably deviate from the ratio of 4:1 expected if they were randomly distributed in the plasmid (Fig. 4b). Surprisingly, all four DPs did not have any microhomology near their endpoints (Supplementary Table 1), despite the fact that near-exact repeats exist in these regions. This striking contrast may indicate that the mechanisms of deletions in different regions of the pQGS plasmid may vary due to how IncQ plasmids replicate (see "Discussion").

Mutant plasmids start as a single copy within one cell, so the dynamics of plasmid replication and segregation affect how much that cell and its progeny benefit from the mutant plasmid and the chances that copies of the mutant plasmid will survive in the cell population[51]. The lag between when a mutant plasmid first appears in a cell and when that cell's descendants realize the full fitness benefit possible from harboring multiple copies of that plasmid is known as its phenotypic delay[52]. We examined phenotypic delay using multilevel stochastic simulations with parameters fit to be consistent with our E. coli evolution experiment (see "Methods"). If SPs replicated more quickly than other plasmid types due to their smaller size, then they would be expected to have a much shorter phenotypic delay. However, models in which we gave SPs an advantage for within-cell replication were not consistent with the experimentally observed numbers of ancestral and SPs coexisting within cells in our experiment and/or the relative fitness of different cell types. Therefore, we gave all plasmids equal rates (chances) of replication in our simulations, which is consistent with a model of plasmid replication in which initiation, rather than elongation or other steps, is rate-limiting.

These simulations predict that SP cells will initially fare better in terms of their fitness before DP cells surpass them (Fig. 4c). Because a one-to-one trade of a copy of the ancestral plasmid for a SP is more beneficial than replacing it with a DP, the model predicts that, if a cell with a SP and a cell with a DP evolved at the same time, the descendants of the SP cell would be more fit, on average, than those of the DP cell for the first 12 generations. This reduced phenotypic delay is expected to give a newly evolved SP a better chance of avoiding stochastic loss. For example, with the daily 1:2000 serial transfer regime of our E. coli experiment, the simulations predict that a new SP has a 17% greater chance than a new DP of establishing a population of descendant cells that is large enough it no longer risks being randomly lost to dilution during transfer.

## Discussion

Acquisition of new antibiotic resistance genes often imposes a fitness burden on a bacterial cell. Thus purifying selection within a bacterial population favors eliminating these and other types of newly acquired accessory genes during times when they are not important for survival or fitness (e.g., when no antibiotic is present). IncQ plasmids have frequently been identified as mobile carriers of antibiotic-resistance genes in clinical environments[21–24]. We found that nonautonomous SPs that delete both accessory genes and genes necessary for plasmid replication frequently evolved in bacterial populations harboring a synthetic IncQ plasmid, both in culture and in an animal host. Because SPs are

molecular parasites that reduce the copy number of full-length plasmids in a cell, SP evolution can rapidly alleviate much of the burden of newly acquired accessory genes without leading to their complete loss from a cell. The improved fitness of cells with SPs makes it more likely that their offspring will survive and maintain copies of these accessory genes on plasmids until a future time when these genes are beneficial rather than burdensome. The ability to transiently evolve SPs creates new pathways favoring preservation versus loss of accessory genes after a plasmid colonizes a new cell (Fig. 5). This evolutionary "safety valve" can prolong the maintenance of accessory genes, such as those encoding antibiotic resistance, as a lineage of IncQ plasmids spreads and persists.

Satellite viruses that require a helper virus to complete their lifecycles have been identified in microbes, plants, and animals[53,54], and how bacterial viruses and their satellites co-evolve has been studied in laboratory evolution experiments[55]. We report that bacterial plasmids can also evolve these types of molecular parasites. Why are IncQ plasmids so prone to SP formation? Nonautonomous "cheater" plasmids like these can only evolve readily if a plasmid encodes factors needed for its replication separately from origin sequences that are sufficient to direct replication. In the pQGS plasmid used in this study and natural IncQ plasmids[33], the origin of replication (oriV) is located at one end of the plasmid backbone, which makes it possible for a single deletion to remove the costly accessory genes and one or more genes encoding replication factors while leaving the origin intact. Although there are some plasmids with replication mechanisms that are independent of any plasmid-encoded proteins (e.g., ColE1 plasmids), most bacterial plasmids do encode at least one protein (usually named Rep) that is needed to initiate replication outside of the origin sequences to which it binds. Thus these plasmids could theoretically evolve or be engineered into SPs, as long as SPs and intact full-length plasmids are able to stably coexist within cells and their progeny.

IncQ plasmids replicate by a strand displacement mechanism and do not encode a partitioning system, so they randomly segregate into daughter cells[25]. RepC has been reported to positively regulate copy number[56], while MobC and MobA suppress it[57]. Despite these regulatory feedbacks, we found that total plasmid copy number was largely unchanged in cells that evolved SPs that inactivated repC in the E. coli experiments. We also observed SPs with a large range of sizes that deleted a variety of combinations of replication and mobilization genes, indicating that many different deletions are compatible with stable maintenance of these SPs. Our modeling indicates that smaller pQGS-derived SPs do not have a significant advantage over full-length plasmids in terms of their replication rate within cells. This could be another reason that it is possible for a cell to maintain enough full-length plasmids such that there is not too great a fitness cost for SP evolution due to some of a cell's progeny inheriting only nonautonomous plasmids and thereby losing plasmid function entirely. It is possible that plasmids from other families, which commonly utilize theta or rolling-circle replication mechanisms, are subject to regulatory controls that make the presence of derived SPs more disruptive to their continued replication and segregation.

The rate at which new SPs are generated during plasmid replication is another important factor that determines whether they will be observed relative to other plasmid fates. We investigated whether there were aspects of SP organization and population dynamics that could favor their evolution in our experiments, especially relative to plasmids that delete burdensome accessory genes but maintain the ability to self-replicate. We found that there are more opportunities for deletions that generate SPs (i.e., mutations that generate them are expected to be more common) and

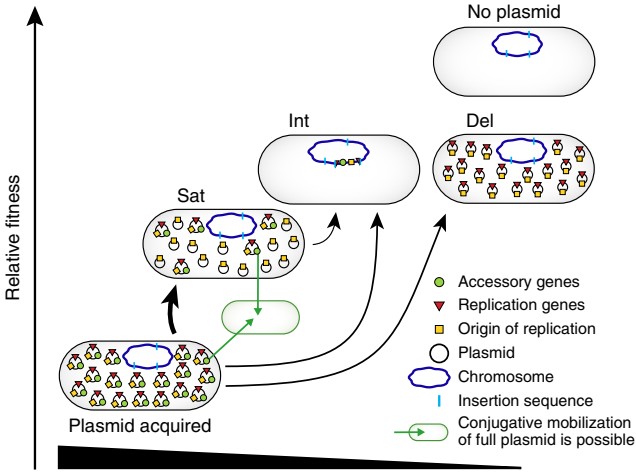

**Fig. 5 Evolutionary pathways after plasmid acquisition.** This model summarizes various evolutionary fates of IncQ plasmids observed in this study and how they may relate to the maintenance of accessory genes on these plasmids in nature. After a new cell acquires an IncQ plasmid, the evolution of nonautonomous satellite plasmids that are dependent on a full-length plasmid in the same cell for replication can alleviate the fitness burden of accessory genes encoded on the plasmid by reducing their copy number (Sat). These cells can still potentially benefit from the accessory genes if the environment changes (e.g., if they provide antibiotic resistance and there is treatment), and full-length plasmids in these cells remain capable of being transferred by conjugation to other cells to spread these accessory genes. Formation of satellite plasmids occurs at a high rate, but this state is likely to be a transient evolutionary intermediate. In experimental populations, cells with satellite plasmids are later displaced by cells that have integrated plasmid sequences into the chromosome via a mechanism involving an insertion sequence (Int) or the evolution of autonomous plasmids that delete just the accessory genes (Del). In the current study, cells had to maintain a gene from the plasmid that is encoded adjacent to the replication genes for survival (which is why no arrows are shown to the "no plasmid" state). Even without this constraint, the ability to evolve satellite plasmids is expected to give more opportunities for preserving the functions of accessory genes encoded on IncQ plasmids as they colonize new bacterial cells and hosts. The fitness and accessory gene copy number scales are not meant to be quantitative; they show only the approximate relative values of these parameters. The sizes of the black arrows for transitions are roughly weighted by the relative rates inferred for these processes from the evolution experiments.

that they are expected to initially give greater fitness benefits to the host cell and its progeny (i.e., they have a reduced phenotypic delay). However, their rather slight advantages in these areas are not enough on their own to explain the prevalence of SPs in our experiments.

Our observation that the deletions which create SPs are flanked by sequence microhomologies but those that generate accessory gene DPs are not suggests that the unusual strand displacement mechanism of IncQ plasmid replication may be a key reason that SPs evolve in our experiments. This mechanism begins with RepC binding to the iteron in *oriV* and disrupting the DNA double helix. Then RepA unwinds the DNA and exposes the *ssiA* and *ssiB* single-strand initiation sites to the primase RepB. RepB primes DNA synthesis in a 5′ to 3′ direction separately on both strands producing large quantities of ssDNA intermediates[25,58]. The mobilization mechanism of IncQ and other plasmids, in which double-stranded plasmid molecules are nicked by the plasmid-encoded relaxase at *oriT* and unwound for conjugative transmission, further contributes to the generation of large

amounts of ssDNA inside of a host cell[28,59,60]. ssDNA is a key player for RecA-independent recombination mediated by annealing of short sequence homologies[61–63]. We hypothesize that the presence of large amounts of ssDNA in cells with the pQGS plasmid may explain why SPs evolve so quickly. Deletions that remove only the accessory genes may occur at a much lower rate and through alternative mechanisms because both of their endpoints are closer together and located on the same arm of the plasmid relative to *oriV*. Similarly, plasmids with theta or rolling-circle replication mechanisms may only rarely evolve SPs.

Formation of SPs is one mechanism that can lead to persistence of accessory genes after they are transferred to a new bacterial host. Another is that the plasmid can become temporarily or permanently single-copy and nonautonomous by integrating into the bacterial chromosome. Integration was a common endpoint in our *E. coli* evolution experiment. In the case that we fully characterized, this process occurred via insertion of an IS*186* copy into the plasmid backbone and recombination with a copy of this transposable element in the genome. IS elements have been broadly reported to facilitate the assimilation of genes carried on incompatible or unstable plasmids into bacterial chromosomes. Similar integration events have also been observed in experiments with different families of conjugative or suicide plasmids in *Pseudomonas putida*[10] and *Yersinia pseudotuberculosis*[64]. In another instance, reversible chromosomal integration mediated by IS elements was reported for a virulence plasmid in *Shigella*[65].

We have shown that a transmissible plasmid possesses an organization and replication mechanism that can favor the rapid evolution of SPs that reduce the copy number of any costly accessory genes that it has spread to a new cell. Awareness of the propensity of IncQ plasmids to evolve SPs is an important consideration when studying the ecological range of this BHR plasmid and when using it to engineer cells. In this latter context, SP evolution could be interpreted as a fortuitous "safety valve" for titrating down toxic or burdensome gene expression or as an unwanted complication that leads to a different effective copy number for engineered DNA sequences than was intended. In summary, our results suggest that SPs may be a common evolutionary intermediate that can stabilize the spread of plasmid-borne accessory genes, including those responsible for antibiotic resistance.

## Methods

***E. coli* evolution experiment**. Plasmid pQGS was constructed out of genetic parts from a Golden Gate toolkit[45]. Briefly, a RSF1010-derived pMMB67EH plasmid (ATCC 37622) backbone part was assembled with a *gfp* optim-1 variant driven by the constitutive PA3 promoter using BsaI assembly. The *aadA* and *lacI* genes were already present in this backbone part. The full sequence of pQGS is available in Genbank (MH423581 [https://www.ncbi.nlm.nih.gov/nuccore/MH423581]). Six individual colonies of *E. coli* BW25113 carrying pQGS were isolated to initiate the evolution experiment. Populations were grown in 10 ml of lysogeny broth (LB) (10 g NaCl, 10 g tryptone, 5 g yeast extract per liter) with 60 μg/ml spectinomycin at 37 °C in 50-ml Erlenmeyer flasks with 200 r.p.m. orbital shaking over a 1-inch diameter unless otherwise noted. Cultures were transferred via a 1:2000 dilution into 10 ml of fresh LB-Spec every day. Samples of each population were periodically archived by adding glycerol as a cryoprotectant and storing at –80 °C.

**Flow cytometry**. Fluorescence was monitored by flow cytometry using a BD LSRFortessa instrument. Cells were counterstained by adding the red membrane dye FM 4–64 (ThermoFisher) to a final concentration of 10 ng/μl to 100 μl of cell culture. The culture was incubated for 10 min at 37 °C in 1.5-ml microcentrifuge tubes with 200 r.p.m. shaking. Then cells were spun down and stored on ice until sample loading. Right before loading, each cell sample was resuspended by adding 1 ml saline and pipetting up and down thoroughly. An aliquot of resuspended cells was diluted 1:20 with saline in 5 ml Falcon round-bottom culture tubes for loading on the flow cytometer. Flow cytometric data files are provided as Supplementary Data 1.

**Quantification of *gfp* and *oriV* copy number by qPCR**. Copy numbers of the *oriV* region and *gfp* gene on the plasmid relative to a chromosomal gene were

determined by qPCR. The chromosomal gene *dapA* was set up as the reference gene (*dapA*-F: 5′-AGCGTCATCATCACCACATC, *dapA*-R: 5′-GTACTTCGGC GATCGTTTCT) to compare to the *gfp* gene (*gfp*-F: GAGGATGGAAGCGTTCA ACTA, *gfp*-R: 5′-GCAGATTGAGTGGACAGGTAA) and *oriV* region (oriV-F: 5′-TCCAGCGTATTTCTGCGG; oriV-R: 5′-GGATAGCTGGTCTATTCGCTG) sequences on the plasmid. The *dapA* standard curve was constructed using genomic DNA isolated from BW25113 using the PureLink Genomic DNA Mini Kit (Invitrogen). Standard curves for *gfp* and *oriV* were constructed using plasmid isolated from the ancestral strain using the PureLink Plasmid DNA Miniprep Kit (Invitrogen). Total DNA amounts were quantified using a Qubit Fluorometer (ThermoFisher).

For each strain of interest, individual colonies for each biological replicate were isolated from revived freezer stocks and grown in 3 ml of LB-Spec in test tubes for 24 h. The cultures were then diluted 1:100 into 10 ml of fresh medium in 50-ml flasks. The cells were harvested at an $OD_{600}$ of ~0.3 and genomic DNA was isolated using the PureLink Genomic DNA Mini Kit (Invitrogen). qPCR was performed using SYBR Green Real Time PCR Master Mix (ThermoFisher) at a final DNA concentration of 0.4 ng/μl using the ViiA 7 Real Time PCR System (ThermoFisher). Amplification conditions were as follows: initial denaturation for 10 min at 95 °C, followed by 40 cycles of denaturation for 15 s at 95 °C, then annealing and extension for 1 min at 60 °C. Three technical replicates (PCR reactions) were conducted for each biological replicate and for each concentration of DNA used to construct the standard curves.

**Detection of SPs and GFP mutations**. Plasmids were isolated from saturated overnight cultures using the PureLink Plasmid DNA Miniprep Kit (Invitrogen). Restriction digests using AvaI and ScaI were performed according to the manu-facturer's instructions (New England Biolabs). SPs arising in evolved populations were detected by PCR using primers P1 (5′-CCGCAGAAATACGCTGGA) and P2 (5′-CAGCGAATAGACCAGCTATCC). Purified amplicons were Sanger sequenced using the primer facing the *gfp* direction. The *gfp* gene was amplified using primers PA3-F (5′-CGGATGACACGAACTCACGA) and GFP-R (5′-GG TTCGTAACATCTCTGTAACTGCT). PCR products were purified and Sanger sequenced using both GFP primers to check for mutations in the *gfp* gene and promoter region.

**Whole-genome sequencing**. Frozen cultures sampled during the evolution experiment were inoculated via a 1:2000 dilution into 10 ml fresh LB medium and cultured for 24 h. DNA was extracted from 1 ml of each resulting culture using the Invitrogen PureLink Genomic DNA Mini Kit (Invitrogen). This DNA was frag-mented using Fragmentase (NEB). Then Illumina sequencing libraries were pre-pared using the KAPA Low Throughput Library Preparation Kit (Roche) according to the manufacturer's instructions, except that all reactions were carried out in one half reaction volumes and both sequencing adapters were modified to include six base barcodes at the beginning of each read. These libraries were sequenced on a MiSeq instrument to generate 150-base paired end reads at the University of Texas at Austin Genome Sequencing and Analysis Facility. Mutations were predicted by using the *breseq* pipeline (version 0.33.2) in polymorphism mode[66,67] to compare sequencing reads to the *E. coli* BW25113 genome (GenBank: CP009273.1) and pQGS plasmid (Genbank:MH423581). FASTQ read files are available from the NCBI Sequence Read Archive (SRP149032).

**Competitive fitness assays**. BW25113 is unable to utilize arabinose (Ara⁻) due to the $\Delta(araD$-$araB)567$ mutation. We created an Ara⁺ variant of BW25113 as a reference strain to compete against Ara⁻ strains from the evolution experiment. To do so, we first inserted a $Spec^R$ marker upstream of the intact *ara* operon in *E. coli* strain MG1655 using Lambda Red homologous recombination[68]. The resulting strain was used as a donor for phage P1 transduction[69] to move the $Spec^R$ marker and the linked *ara* operon into BW25113 to create the reference strain. The plasmid integration strain (Int2) was constructed by P1 transduction of the single-copy integration of the plasmid in the chromosome of strain B251 into BW25113. The *gfp* deletion strain (Del6) was made by transforming a plasmid from an evolved strain isolated from population 6 at day 20 into BW25113. Strains carrying both the pQGS plasmid and a SP (Sat2 and Sat6) were made by transforming the purified SPs from strains B253 and B653, respectively, into the ancestral BW25113 cells containing pQGS. When constructing the SP strains, all cells in the initial transformation reaction were cultured through two growth cycles in LB-Spec to enrich for cells that acquired the SP. Then these populations were plated on LB-Spec agar and colonies were screened for the presence of SP by PCR and Sanger sequencing to identify the clones used in the fitness assays.

For each competition experiment comparing two strains, six colonies of the strain to be tested and six of the reference strain were picked to initiate separate overnight test tube cultures in 3 ml of LB (for the BW25113 versus reference strain competition) or LB-Spec (for all other competitions). These cultures were diluted 2000-fold into 10 ml fresh medium in 50-ml Erlenmeyer flasks and incubated for 24 h. These pre-conditioned cultures of each strain of interest were mixed with the reference strain in triplicate at volume ratios ranging from 1:1 to 28:1 depending on the relative fitness of the two strains, while always maintaining an overall culture dilution of 2000-fold, into 10 ml of fresh medium to begin the competition. The

ratio of each strain before and after 24 h of growth was determined by plating dilutions on tetrazolium arabinose agar plates. Relative fitness values were calculated for each of the total of 18 replicate competition assays for a pair of strains as previously described[70].

**Bee gut symbiont experiments**. Plasmid pQGS was transferred into *S. alvi* wkB2 via conjugation[45]. The donor strain (*E. coli* MFDpir transformed with pQGS) was grown in 3 ml of LB supplemented with 0.3 mM diaminopimelic acid (DAP) in test tubes for 24 h at 37 °C with 200 r.p.m. shaking. The recipient strain (*S. alvi* wkB2) was grown on Heart Infusion Agar (HIA) (Oxoid) for 2 days. The donor and recipient cells were harvested by centrifugation or scraping the plate, washed in 1 ml of phosphate-buffered saline (PBS) separately, spun down, and resuspended in 1 ml PBS. These two suspensions were mixed in a 1:1 $OD_{600}$ ratio and spotted onto HIA supplemented with 0.3 mM DAP. This plate was incubated at 35 °C under 5% $CO_2$ overnight. Cells were scraped from the plate and washed twice in 1 ml PBS to remove residual DAP. Then 50 μl of a dilution of the cells was spread onto HIA supplemented with 30 μg/ml Spec and incubated at 35 °C under 5% $CO_2$ for 2 days. Candidate transconjugants were passaged again on selective agar and confirmed by PCR amplification and Sanger sequencing of the *gfp* gene sequence on the plasmid. For the in vitro evolution experiment, five populations were started from single colonies by inoculating them into Insectagro DS2 medium (Corning) amended with 30 μg/ml Spec (DS2-Spec). *S. alvi* was cultivated in 6 ml of this medium in culture tubes at 35 °C under 5% $CO_2$ without shaking. At 4-day intervals, 6 μl of each culture was transferred into 6 ml of fresh DS2-Spec and checked for the presence of SPs by PCR and Sanger sequencing.

For the in vivo evolution experiment, microbiota-free bees were obtained by removing late-stage pupae manually from brood frames and placing them in sterile plastic bins[48]. The pupae emerged in an incubator at 35 °C and humidity of 75%. Newly emerged bees were kept in cup cages with sterilized sucrose syrup (0.5 M) and bee bread. For inoculating these bees, *S. alvi* wkB2 cells containing the pQGS plasmid cultured on DS2-Spec agar were first suspended in 1 × PBS to an $OD_{600}$ of 1.0. Then batches of 20–25 bees were placed into a 50-ml conical tube, and 50 μl of sucrose syrup was added. The tube was rotated gently to coat the surfaces of bees with the syrup. Finally, 50 μl of the *S. alvi* wkB2/pQGS suspension (~5 × 10⁷ cells) was added to the tube, and it was rotated again. *S. alvi* colonizes the guts of these surface-coated bees as a result of autogrooming and allogrooming.

Inoculated bees were reared in cup cages. Five of these enclosures were set up in each of the two different experiments with at least 12 bees in each cage. Bees were fed sucrose syrup with spectinomycin (60 μg/ml) throughout the experiment. Bee guts were dissected 24 h after inoculation and examined for the presence of wkB2/pQGS by fluorescence microscopy. To measure *S. alvi* wkB2/pQGS colonization levels, bee guts were dissected and homogenized on days 3 and 4 of the experiment. Serial dilutions of these samples in PBS were plated on HIA and incubated at 35 °C under 5% $CO_2$ for 1–2 days before counting CFUs to determine the number of bacteria per gut[71].

To characterize SP evolution, one bee gut from each cup was dissected every day until 4 days after inoculation. Bee guts were homogenized in 728 μl of cetyltrimethylammonium bromide (CTAB) buffer with 20 μl Proteinase K solution (20 mg/ml). DNA was then extracted using a bead-beating method[72]. Briefly, dissected guts were placed in tubes containing ~0.5 ml of 0.01-mm silica zirconia beads (BioSpec Products), 728 μl CTAB, 2 μl 2-mercaptoethanol (Sigma), and 20 μl of 20 mg/ml proteinase K (Sigma). The tubes were processed in a multi-sample bead beater (BioSpec Products). Next, samples were incubated at 56 °C overnight, 5 μl of RNase A (Sigma) was added, and the tubes were vortexed briefly and placed at 37 °C for 1 h. After this, samples were combined with 750 μl phenol:chloroform:isoamyl alcohol (25:24:1) (Ambion), shaken for 30 s, and placed on ice for 2 min. Then the tubes were centrifuged at 4 °C, and the aqueous phase was ethanol precipitated, washed, and air dried prior to being resuspended in 50 μl of nuclease-free water. SPs in these samples were detected by PCR and characterized by Sanger sequencing.

**Repeat analysis**. We created a Python script to enumerate microhomologies (short, near-perfect repeats) in pQGS that could mediate deletions leading to SP formation or accessory gene loss. We ran this code with settings that identified repeats that were at least 7 bases long, included at most one inserted or deleted base (indel), had at most five total mismatches (including indels), and with at least 75% sequence identity. When multiple valid alignments overlapped, only one was counted, giving precedence to the longer one, and then to the one with fewer mismatches.

**Plasmid evolution simulations**. We implemented multilevel stochastic simula-tions that replicate cells and plasmids using Monte Carlo methods in Python. We established parameters that fit the observed numbers and types of plasmids typical for each cell type to its fitness by first assuming a linear cost for expression of each protein from the plasmid[73,74]. With 18 total plasmids per cell, an additive fitness model with a cost of 3.06% per full-length plasmid and 0.50% per DP in a cell matches the experimentally observed fitness values for cells containing solely ancestral or solely DPs. A model in which SPs have no fitness cost and the col-lection of plasmids in each cell is exactly doubled and randomly but evenly assorted to daughter cells leads to an equilibrium level of 5.2 ancestral plasmids and 12.8 SPs per cell, on average. This is close to the ratio of the two types observed for several

evolved strains. Note that, instead of the expected relative fitness value of 0.841 for the cost of those 5.2 full-length plasmids, loss of all ancestral plasmids from some offspring due to segregation causes this population of cells to realize a relative fitness of just 0.77, which also closely matches the experiments. Alternative plasmid replication and segregation models that introduced more complexity and randomness gave lower fitness values and/or prevalence of SPs in cells than we observed. For example, we tested allowing plasmids to replicate via their own stochastic growth process and unevenly segregating plasmids between daughter cells.

The stochastic simulations track a collection of cells of interest that may contain different numbers of each plasmid type as they compete against a background of cells with a fixed fitness value and the cell population is transferred via serial dilution. We used them in three ways: (1) To determine the equilibrium fitness value of each cell type and equilibrium number of SPs per cell, we tuned the fitness of the background population in the model until a population seeded with 10,000 initial cells of the type of interest were able to maintain a stable population size over 500 generations of growth with 2-fold serial dilutions. (2) We examined the phenotypic delay of different plasmid mutations by starting 200 simulated populations with 20,000 cells containing 1 copy of either a SP or a DP and 17 copies of the ancestral plasmid and propagating them for 50 generations with 1.072-fold dilutions in a background population of cells containing only the ancestor plasmid. The realized fitness of mutant subpopulation at a given time point was calculated from the number of doublings it achieved relative to the background population within this time period. (3) To measure the chance that a new cell with a SP or DP would successfully escape dilution and establish under the conditions of our evolution experiment, we started 1,000,000 simulations, each with one cell containing one mutant plasmid that arose at a random cell division within a daily growth cycle with a 2000-fold transfer dilution. Then we followed the cell number of this population competing against a background of cells with only the ancestor plasmid until cells with the mutant plasmid either went extinct or reached a population size that was 20-fold higher than the dilution factor at the end of a growth cycle. The chance that a new mutant will establish is the number of events of the latter type divided by the total number of trials.

**Reporting summary**. Further information on research design is available in the Nature Research Reporting Summary linked to this article.

## Data availability
Genome sequencing data are available from the NCBI Sequence Read Archive (SRP149032 [https://www.ncbi.nlm.nih.gov/sra/?term=SRP149032]). The full sequence of pQGS is available in Genbank (MH423581 [https://www.ncbi.nlm.nih.gov/nuccore/MH423581]). The *E. coli* BW25113 genome analyzed for comparison in this study is available in GenBank (CP009273.1 [https://www.ncbi.nlm.nih.gov/nuccore/CP009273.1]). Flow cytometric data are available in Supplementary Data 1. All other datasets generated and/or analyzed during the current study are available from the corresponding author on reasonable request. The source data underlying Fig. 1c–e, 2a, b, and e–h, 3a, c, and 4b, c, and Supplementary Figs. 1b–c, 2, 4a, and 5 are provided as a Source Data file.

## Code availability
Python and R scripts for the repeat analysis and population genetics simulations are freely available online (https://github.com/barricklab/satellite-plasmid).

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

## Acknowledgements

We thank Nancy Moran for the use of her laboratory's facilities for honey bee experiments. Sean Leonard and other researchers in the Barrick and Moran laboratories provided helpful feedback on this project. The authors acknowledge the Texas Advanced Computing Center (TACC) for providing high-performance computing resources. This work was funded by the Defense Advanced Research Projects Agency (HR0011-15-C0095) and the National Science Foundation (CBET-1554179). The funders had no role in study design, data collection and interpretation, or the decision to submit the work for publication.

## Author contributions

X.Z. and J.E.B. conceived and designed the experiments. X.Z. and H.Z. performed the experiments. S.J.G. and J.E.B. performed the simulations. X.Z, D.E.D., and J.E.B, analyzed data. X.Z. and J.E.B. wrote the paper.

## Competing interests

The authors declare no competing interests.
