## [Peer Review File · Nature Communications]

Reviewers' Comments:

Reviewer #1:

Remarks to the Author:

In this work, the authors explore the evolution of IncQ plasmid-carrying populations of *E. coli* and *Snodgrassella alvi*. Selection for plasmid-mediated antibiotic resistance drives compensatory evolution ultimately leading to plasmid mutation (deletion of costly *gfp* gene) or plasmid insertion into the chromosome assisted by an IS element. Interestingly, an intermediate state arises where large deletions in the plasmid produce defective plasmid molecules, which lose accessory genes and replication genes but not the *oriV* (Satellite plasmids, SP). These molecules depend therefore on wt copies of the plasmid to allow their replication but, at the same time, carrying a fraction of the total cellular plasmid copy number via SPs reduces the global burden imposed by *gfp*, while maintaining copies of the resistance gene under selection. This effect is driven by plasmid copy number control "counting" both wt plasmids and satellite plasmids as the same thing, and therefore maintaining the total count (addition) of both elements at a constant level. This intermediate state emerges in multiple replicates, in the two species and both in vitro and in vivo. To my knowledge, this interesting strategy to reduce plasmid cost had not been reported before. The authors show that despite the instability of SPs, the high accessibility to mutations producing these genotypes, which is facilitated by the nature of plasmid replication mechanism, promotes the pervasive emergence of SPs in all populations. In my opinion, this is an interesting and elegant work that uncovers a new mode of plasmid evolution that may be common and relevant in nature.

Major comments:

-My only substantial suggestion to the authors would be to try to look for signatures of SPs in databases. It would be interesting to look for bacterial genomes isolated from single clones carrying IncQ plasmids in databases, and then try to assemble the plasmids and look for differences in coverage across the sequence that may indicate the presence of SPs. If this analysis is successful it could indicate that SPs are frequent in nature.

Minor comments:

-lines 61-63, there are also examples of compensatory evolution arising in the absence of selection in experiments, although it is true it usually takes a while.

-line 67, it may be worth defining conjugative (self-transferable), mobilizable and non-transferable plasmids at this point.

Alvaro San Millan

Reviewer #2:

Remarks to the Author:

The paper of Zhang et al. investigates the evolution of an IncQ RSF1010 plasmid derivative (pQGS) carrying multiple accessory genes in *Escherichia coli*. The authors conduct an evolution experiment in antibiotics using six replicates populations in a serial transfer system. The experiment reveals a decrease in the plasmid copy number (in some populations) that was due to the formation of satellite plasmids. The satellite plasmids are characterized by the loss of most of their genetic load including the antibiotic resistance gene, *gfp* and most of the replication machinery and were thus dependent on other plasmid copies for replication. The authors propose that satellite plasmids may be advantageous as the lower genetic load and copy number in turn lowers the fitness burden for the host cell. Indeed, fitness experiments validate that strains with satellite plasmid have a lower fitness impact than the plasmid ancestor. However, the authors discover that at the end of the evolution experiment, all satellite plasmid disappeared and instead deletion plasmid (lost *gfp*) or chromosomal plasmid integration were prevalent. Overall, the study displays very interesting novel results about the

evolutionary dynamics of plasmids and nicely shows that how plasmid character (replicon type, multi-copy, small) may enable distinct evolutionary solutions to maintain antibiotic resistance. I expect that this study would be of high interest to microbiologists, evolutionary biologists and scientists studying the spread of antibiotics resistance. I have several comments/suggestions that could further improve this manuscript.

Specifically, the study needs some clarifications: the authors should explain whether the observed scenario is expected in nature or a result of the artificial plasmid design (GFP leads to formation of satellite plasmids?). In addition, the authors should emphasize how their findings might be relevant for the evolutionary dynamics of bacteria and plasmids in nature. In the end, the plasmid accessory gene deletions (or integration) are favored over satellite plasmids. Thus, satellite plasmids are very much transient and in the end fully functional plasmids (or integration) is selected. A consequence the authors should discuss in more detail. Lastly, the justification for the introduction of the plasmid into the honey bee gut is not clear.

Comments:

1. Plasmid copy numbers are different between populations (Fig. 1C). How do the authors explain the increase in copy number in some populations (e.g., B453)?
2. The authors mention that the plasmid-type is 'hard to visualize', nevertheless, they manage to extract the plasmids. Circularized satellite plasmids should be shown, to be able to definitely exclude replication rearrangements/intermediates.
3. The copy number for GFP is missing for the sequencing data (Fig. 2F, F). In addition, the authors should indicate if the sequencing validated the qPCR measurements.
4. The satellite plasmids are different in the two sequenced populations (Fig 2 A,B). Specific satellite types seem to be dominant in distinct populations. It is not clear what's the reason for the difference and how this might influence the endpoint state of integration or deletion plasmid. It is interesting that in Sat6 and the deletion plasmid coexist for a while in B6.
5. In the fitness experiments, the deletion plasmid seems to be equally advantageous than chromosomal integration. However, the integration seems to be present in the population B6 during the experiment. Thus, the deletion plasmid may be favored over chromosomal integration. The authors should explain that (in the results or discussion). Same line of thoughts: Do the authors think the deletion plasmid might occur in the other populations at a later stage? And why is the deletion observable in the *gfp* data but not in the sequencing reads for B2 (Fig. 2 C and E)?
6. If I get it right, the objectives of the honey-bee gut experiments are to demonstrate the occurrence of SPs also in other systems (i.e., in 'nature'). At the moment, this comes however quite as a surprise in the manuscript. The authors might want to elaborate on: what is the justification to introduce the plasmid-carrying bacteria into the bee? Is it a specific environment where IncQ plasmids are known to occur?
7. The number of replicates tested $n=xx$ is sometimes missing and should be mentioned in the results or the figure legends (e.g. fitness experiments).
8. Sequencing results: a coverage plot of specific satellite plasmids would be a further proof for their

existence. Can they be assembled from the sequencing reads?

9. Overall, satellite plasmids seem to be transient while other solutions are favored over time. This is not in agreement with the title (and overall statements) as the accessory gene *gfp* is lost in some cases. The satellite plasmids may be selected because they reduce the fitness burden until the deletion/integration is fixed in the lineage.

Reviewer #3:

Remarks to the Author:

In this manuscript, Zhang and colleagues investigate an intriguing evolutionary trajectory of a new plasmid-bacterial association. Using an elegant combination of experimental evolution, molecular biology, whole genome sequencing, genetic reconstruction, and stochastic modelling, the authors describe how the synthetic IncQ plasmid pQGS imposed a significant cost on the bacteria that host it, but these costs were driven down by the emergence of satellite plasmids that parasitise and compete with the original plasmid, suppressing copy number and thus cost.

Overall I thought that the work was well-conducted and explained, with clear figures. Though satellite elements have been reported before, this study provides interesting context to their evolution, and thus will be of interest to a broad range of readers.

Main comments and suggestions:

1. My understanding is that in all populations, satellite plasmids emerged, but were transient. The ultimate fate of the plasmid was to either become integrated or to lose the accessory gene(s). The role of the satellite plasmids in stabilising accessory genes therefore is essentially to provide an easily accessed, intermediate step on the fitness landscape that results in accessory genes being maintained longer than would be expected if the population immediately evolved the deletion plasmid. This idea — that plasmid evolutionary trajectory is determined partially by plasmid-plasmid competition allowing satellite plasmids to beat deletions in the short term — is really interesting. However the transience of the satellite plasmids in host-plasmid evolution should be made clearer, at least in the abstract and in the summary. It currently reads as if satellite plasmids are relatively stable long-term (e.g. line 24-24 "the evolution of satellite plasmids relative to other plasmid fates") which is confusing. A diagram showing the steps involved, the incremental fitness changes, the long-term outcomes, and the role of the satellite plasmid in this process, e.g. as a supplementary figure similar to Figure 7 in Cury et al. doi: 10.1093/molbev/msy123 or <https://natureecoevocommunity.nature.com/users/55681-michael-bottery/posts/18889-adapting-to-resistance> might also help.

2. Again, the authors propose that satellite plasmid generation is a widespread mechanism in plasmid (particularly/exclusively IncQ plasmid) evolution for alleviating costs associated with acquisition of new accessory genes (lines 420-421). However this seems to be in contrast to the statement that IncQ plasmids are evolutionarily stable (lines 89-90). I assume that this is because satellite plasmids are short-term/evolutionary dead ends: they are cheaper than ancestors, but are doubly vulnerable to loss by segregation as they are also lost if the co-resident ancestral plasmid they are parasitising is lost? This could be made clearer.

3. Is there any evidence of satellite plasmids in natural isolates? Could publicly available sequencing data be used to look for variance in coverage across IncQ plasmids?

4. Figure S2. The authors might like to consider ordering the fragments by population rather than by

size, so that the within- and between-population diversity can be assessed, as well as parallelism between clones from different populations.

5. It is not always clear whether PCR and sequencing is performed on individual clones or on the whole population. E.g. line 171-172 I believe refers to the PCR assay conducted on the whole populations — this should be specified. In addition, the authors should specify how many clonal isolates were sequenced per population, either here or in the legend to Figure S2.

6. Figure S2 is titled "Maps of evolved plasmids found in all experiments", which is confusing — I don't think these were found in all the experiments. In addition, I don't know how the authors can be sure they have exhaustively mapped all the evolved plasmids across all experiments. Perhaps "Maps of evolved plasmids identified in isolate clones" might be a better title?

7. Figure 2 and Figure S3 show different times of sampling across the panels, which is confusing when trying to compare assays between timepoints. In addition the x-axes are not always linear. This should be explained or at least mentioned in the legends.

8. Line 195. It would help the reader to specify here that WGS was performed across the course of the experiment and to give the number of timepoints sequenced.

9. The role of the IS sequence in integrating the plasmid is interesting and could bear a further sentence or two of discussion, e.g. how IS elements might facilitate capture/domestication of incompatible or otherwise unstable plasmids. See e.g. Lesic et al. doi: 10.1371/journal.pgen.1002529

10. Line 212-216 — could the hypothesis of multiple integrants be tested by qPCR on clones? At the moment this model is somewhat speculative.

11. Line 210 describes a PCR technique to test for integration. Ideally this should be extended to the other populations to support the flow cytometry data.

12. The authors expand the scope of their study to investigate plasmid evolution in a more natural system (honey bee gut), making their conclusions more general, which is great. However all experiments are still performed with an artificial plasmid. Could satellite plasmid evolution be a feature of this artificial plasmid rather than of IncQ plasmids in general? A word or two on this would help readers understand the generality of the authors' findings.

13. Though they largely correlate, there are some inconsistencies between the different methods of assessing plasmid presence (PCR/FC/WGS) which should be mentioned/explained. For example, how do the authors explain the persistence of the band through to day 18 in 2A with the lack of 'satellite plasmid' fluorescence in 2C? The lack of a 'del' band in 2A given the low levels of del FC in 2C? Or the lack of 'satellite plasmid' fluorescence in 2D given the band in 2B?

14. Is it possible for satellite plasmids to parasitise deletion plasmids? Do the authors detect this at all?

15. The authors should consider including a few words about parasitic/satellite elements in other systems in their discussion, e.g. in bacteriophage, see for example Frígols et al. 2015 doi: 10.1371/journal.pgen.1005609.

Thank you for the helpful comments and the opportunity to improve our manuscript. Our revisions in response to the comments and related points of discussion are described below.

Reviewers' comments:

Reviewer #1 (Remarks to the Author):

In this work, the authors explore the evolution of IncQ plasmid-carrying populations of *E. coli* and *Snodgrassella alvi*. Selection for plasmid-mediated antibiotic resistance drives compensatory evolution ultimately leading to plasmid mutation (deletion of costly *gfp* gene) or plasmid insertion into the chromosome assisted by an IS element. Interestingly, an intermediate state arises where large deletions in the plasmid produce defective plasmid molecules, which lose accessory genes and replication genes but not the *oriV* (Satellite plasmids, SP). These molecules depend therefore on wt copies of the plasmid to allow their replication but, at the same time, carrying a fraction of the total cellular plasmid copy number via SPs reduces the global burden imposed by *gfp*, while maintaining copies of the resistance gene under selection. This effect is driven by plasmid copy number control “counting” both wt plasmids and satellite plasmids as the same thing, and therefore maintaining the total count (addition) of both elements at a constant level. This intermediate state emerges in multiple replicates, in the two species and both in vitro and in vivo. To my knowledge, this interesting strategy to reduce plasmid cost had not been reported before. The authors show that despite the instability of SPs, the high accessibility to mutations producing these genotypes, which is facilitated by the nature of plasmid replication mechanism, promotes the pervasive emergence of SPs in all populations. In my opinion, this is an interesting and elegant work that uncovers a new mode of plasmid evolution that may be common and relevant in nature.

Major comments:

-My only substantial suggestion to the authors would be to try to look for signatures of SPs in databases. It would be interesting to look for bacterial genomes isolated from single clones carrying IncQ plasmids in databases, and then try to assemble the plasmids and look for differences in coverage across the sequence that may indicate the presence of SPs. If this analysis is successful it could indicate that SPs are frequent in nature.

Reply: This is a great suggestion. We also thought about looking for evidence of satellite plasmid evolution in sequence databases and attempted this analysis prior to submitting our manuscript. Unfortunately, there is very little deep sequencing data available for IncQ plasmids.

We found one study that deposited Illumina reads in the European Nucleotide Archive (Project: PRJEB18259) from sequencing putative *E. coli* transconjugants which might have acquired IncQ plasmids from soil bacteria [1]. However, analyzing these reads with PlasmidFinder [2] did not detect any plasmid origin sequences. We also analyzed Illumina whole-plasmid sequencing data that we obtained from Addgene for nine IncQ plasmids that they distribute (Accessions: 85581, 89476, 90102, 90103, 90104, 103856, 103858, 103860, 106395). For these samples, we followed the analysis procedure we used in our study: we mapped all reads to the Addgene assemblies of the full-length plasmids using *breseq*. We did not find any split-read alignments or differences in

read-depth coverage that indicated satellite plasmids might be present in a mixture alongside full-length plasmids in these samples. Overall, given how little sequencing data is available, we think these inconclusive results are probably not worth mentioning in the manuscript.

1. Garbisu, C., Garaiurrebaso, O., Lanzén, A., Álvarez-Rodríguez, I., Arana, L., Blanco, F., Smalla, K., Grohmann, E., Alkorta, I. (2018) Mobile genetic elements and antibiotic resistance in mine soil amended with organic wastes. *Sci. Total Environ.* **621**: 725–733. <https://doi.org/10.1016/j.scitotenv.2017.11.221>

2. Carattoli, A., Zankari, E., Garcíá-Fernández, A., Larsen, M. V., Lund, O., Villa, L., Aarestrup, F. M., Hasman, H. (2014) *In silico* detection and typing of plasmids using plasmidfinder and plasmid multilocus sequence typing. *Antimicrob. Agents Chemother.* **58**: 3895–3903. <https://doi.org/10.1128/AAC.02412-14>

Minor comments:

-lines 61-63, there are also examples of compensatory evolution arising in the absence of selection in experiments, although it is true it usually takes a while.

Reply: Thank you for suggesting this correction. Yes, compensation can sometimes arise in the absence of selection. One example is the *gacA* and *gacS* mutations that arise when propagating *Pseudomonas fluorescens* carrying pQBR103 in the 0 μM HgCl_2 treatment in reference 13 (Harrison et al. 2015; <https://doi.org/10.1016/j.cub.2015.06.024>). We now mention that there are sometimes exceptions, citing this study in the revised manuscript (Lines 80-84):

However, it usually takes a long time—several hundred cell generations—for compensatory mutations to arise when these mechanisms have been directly observed in laboratory populations of microbes (12, 18, 19). With some exceptions (e.g., 13), these experiments must generally apply constant selection for plasmid function to observe compensatory evolution. These long timescales and stringent conditions may be unrealistic in most natural environments.

-line 67, it may be worth defining conjugative (self-transferable), mobilizable and non-transferable plasmids at this point.

Reply: Yes, that's a helpful distinction to make more clearly. We edited to state that IncQ plasmids are mobilizable rather than self-transmissible on Lines 99-101:

All IncQ plasmids encode multiple mobilization genes, including *mobA*, *mobB*, *mobC*, *mobD*, *mobE*, and the transmission initiation site, *oriT* (24). They are mobilizable but not self-transmissible because they rely on mating pilus formation by a conjugative plasmid (e.g., an IncP or IncN plasmid) or by the host cell for DNA transfer (29, 30).

Reviewer #2 (Remarks to the Author):

The paper of Zhang et al. investigates the evolution of an IncQ RSF1010 plasmid derivative (pQGS) carrying multiple accessory genes in *Escherichia coli*. The authors conduct an evolution experiment in antibiotics using six replicates populations in a serial transfer system. The experiment reveals a decrease in the plasmid copy number (in some populations) that was due to the formation of satellite plasmids. The satellite plasmids are characterized by the loss of most of their genetic load including the antibiotic resistance gene, *gfp* and most of the replication machinery and were thus dependent on other plasmid copies for replication. The authors propose that satellite plasmids may be advantageous as the lower genetic load and copy number in turn lowers the fitness burden for the host cell. Indeed, fitness experiments validate that strains with satellite plasmid have a lower fitness impact than the plasmid ancestor. However, the authors discover that at the end of the evolution experiment, all satellite plasmid disappeared and instead deletion plasmid (lost *gfp*) or chromosomal plasmid integration were prevalent. Overall, the study displays very interesting novel results about the evolutionary dynamics of plasmids and nicely shows that how plasmid character (replicon type, multi-copy, small) may enable distinct evolutionary solutions to maintain antibiotic resistance. I expect that this study would be of high interest to microbiologists, evolutionary biologists and scientists studying the spread of antibiotics resistance. I have several comments/suggestions that could further improve this manuscript.

Specifically, the study needs some clarifications: the authors should explain whether the observed scenario is expected in nature or a result of the artificial plasmid design (GFP leads to formation of satellite plasmids?). In addition, the authors should emphasize how their findings might be relevant for the evolutionary dynamics of bacteria and plasmids in nature.

Reply: Satellite plasmids like the ones we observed must evolve in nature at times. IncQ plasmids often carry multiple cargo genes. Many of these will be burdensome to maintain, and only some of them will benefit the host cell at any given time. These cargo genes are usually inserted at same place in the RSF1010 backbone at which the *gfp*, and *lacI* genes are inserted into pQGS. We added these details to and explain further the relationship between pQGS and a natural IncQ plasmid with accessory genes in the first paragraph of the Results on lines 155-165:

pQGS has three genes inserted into the RSF1010 backbone at a site where accessory genes are found in natural IncQ plasmids (32). ... For purposes of comparing our experiments to events that may happen in IncQ plasmids evolving in nature, *aadA* serves as a gene that enables cells that acquire the plasmid to survive and flourish immediately, in their current environment.

Therefore, similar deletions in natural plasmids are expected to form satellite plasmids that benefit the fitness of a host cell. We emphasize throughout the manuscript that we think the primary effect of forming satellite plasmids on the evolutionary dynamics of these plasmids and their bacterial hosts is in how it can enable unselected accessory genes to be maintained, not indefinitely, but for longer and in more copies than would be the case without this pathway.

In the end, the plasmid accessory gene deletions (or integration) are favored over satellite plasmids. Thus, satellite plasmids are very much transient and, in the end, fully functional plasmids (or integration) is selected. A consequence the authors should discuss in more detail.

Reply: Please see the reply to **Comment #9** below.

Lastly, the justification for the introduction of the plasmid into the honey bee gut is not clear.

Reply: Please see the reply to **Comment #6** below.

Comments:

1. Plasmid copy numbers are different between populations (Fig. 1C). How do the authors explain the increase in copy number in some populations (e.g., B453)?

Reply: Regulation of IncQ plasmid replication is complex enough that it is not easy to predict how the evolution of satellite plasmids or deletion plasmids will affect copy number. IncQ copy number is determined by interactions between several regulatory circuits (Meyer 2009; <https://doi.org/10.1016/j.plasmid.2009.05.001>). Initiation of replication is controlled by plasmid encoded proteins—including RepB (MobA), RepB', RepA and RepC—which are essential for replication and mobilization. Meanwhile, these proteins regulate a cluster of promoters that their own genes are transcribed from (Frey et al. 1992; [https://doi.org/10.1016/0378-1119\(92\)90675-F](https://doi.org/10.1016/0378-1119(92)90675-F)). In addition, RepF and an antisense RNA are also involved in copy number regulation.

In the case of isolate B453, a satellite plasmid pB453 that is lacking *repF*, *repA* and *repC* evolved (**Fig. S2**). RepF represses the production of RepC. RepC overexpression can increase plasmid copy number several-fold (Haring et al. 1985; <https://doi.org/10.1073/pnas.82.18.6090>). Thus, it is possible that removing *repF* from the satellite plasmid results in less negative regulation of *repC* than normal and an overall increase in its expression and thereby total plasmid copy number in these cells. One might contrast this with B253 and B353 in which the pB253 and pB353 satellite plasmids still delete *repA* and *repC* but retain *repF* (**Fig. S2**). In these cells, the overall copy number (*oriV*) appears to be a bit lower than it is for cells that have only the ancestral plasmid. The error on all of these measurements, but especially B453, is relatively high: the error bars are standard deviations in **Fig. 1c**. The revised version of this figure shows the separate data points for each qPCR biological replicate. There is also the possibility that there are integrations of plasmids in some of these cells (B454, for example), which further muddies the picture. For these reasons, we hesitate to comment further on the possible reasons for changes in copy number in the discussion of these results.

2. The authors mention that the plasmid-type is ‘hard to visualize’, nevertheless, they manage to extract the plasmids. Circularized satellite plasmids should be shown, to be able to definitely exclude replication rearrangements/intermediates.

Reply: This is a great suggestion. We now show gel electrophoresis of the plasmid mixtures isolated from cells with satellite plasmids before and after cleavage with a one-cutter restriction

enzyme in a new figure (new **Fig. S1**, which is reproduced on the next page). The results are in agreement with the PCR assay. They indicate that satellite plasmids are circular DNA molecules that exist independently from the ancestral plasmid and the chromosome in these cells.

Fig. S1 Direct visualization of satellite plasmids in evolved *E. coli* isolates. **a** Map of pQGS plasmid showing the single site cleaved by the restriction enzyme *ScaI* and the regions that are preserved in the satellite plasmids that are present in evolved strains B253, B353, and B653. **b** Total plasmid DNA isolated from each of these strains separated by gel electrophoresis. **c** Total plasmid DNA isolated from each of these strains, digested with *ScaI*, and then separated by gel electrophoresis. After the circular pQGS ancestor plasmid (Anc) is linearized by *ScaI*, the mobility of its band decreases, and it runs at the expected size of 9.3 kb relative to the DNA ladder. Similarly, the mobility of the satellite plasmid bands (Sat) decreases upon cleavage, which is consistent with the satellite plasmids being maintained as circular DNA molecules in these cells. After linearization, the satellite plasmid bands run at sizes that agree with the results of sequencing PCR products to determine which regions of the ancestral plasmid were deleted in each one. Source data are provided as a Source Data file.

We added a description of the data shown in **Fig. S1** in the main text on lines 214-221:

We further confirmed that circular satellite plasmid molecules form in these cells and are maintained alongside copies of the full-length ancestral plasmid by directly visualizing total plasmid DNA purified from B253, B353, and B653 cells by electrophoresis before and after linearizing the DNA molecules via cleavage at a single site with the *ScaI* restriction enzyme (**Fig. S1**).

3. The copy number for GFP is missing for the sequencing data (Fig. 2F, F). In addition, the authors should indicate if the sequencing validated the qPCR measurements.

Reply: In our experiments, all plasmids that lost *gfp* function also deleted the entire *lacI* gene. However, the *gfp* gene is only partially removed in the evolved “deletion plasmids” that lost only the accessory genes (see **Fig. S3**). Therefore, we use *lacI* gene copy number in **Fig. 2e–f** as the best way of reproducibly measuring *intact gfp* gene copy number across all plasmid fates. Thank you for pointing out this confusing labeling. We changed the label for this series on the graph to “*lacI (gfp)*” and added an explanation about how these are linked to the caption:

Coverage of the *lacI* gene was used to monitor the copy number of intact *gfp* gene copies because *lacI* is completely deleted in all evolved plasmids, but *gfp* is only partially deleted in some plasmids.

The qPCR data is for individual clonal isolates and the sequencing data is for mixed evolved populations, so they cannot be directly compared. One can see that cells containing satellite plasmids like B253 reach a high frequency on day 4 in population B2, when there is a clear difference in *oriV* and *lacI (gfp)* copy number in **Fig. 2e**. The dynamics become more complicated to detect/interpret from the sequencing data when the integration plasmid evolves.

4. The satellite plasmids are different in the two sequenced populations (Fig 2 A, B). Specific satellite types seem to be dominant in distinct populations. It is not clear what’s the reason for the difference and how this might influence the endpoint state of integration or deletion plasmid. It is interesting that in Sat6 and the deletion plasmid coexist for a while in B6.

Reply: Yes, satellite plasmids with different deletions arose and dominated in different populations. However, this variation is expected in an evolution experiment due purely to chance. There will be stochastic differences in which satellite plasmids evolve in a population, how early they arise relative to one another, how lucky they get at surviving segregation into daughter cells, whether the cells they are in remain after each daily 1:1000 transfer dilution, etc. In evolution experiments like this one, these random differences typically compound to lead to only one or maybe a handful of mutants reaching a high enough frequency for detection (>0.2%) versus remaining below this threshold, even if all of them have very similar fitness benefits.

It would be interesting if the type of satellite plasmid present in the population affected which plasmid fate ultimately prevailed: deletion versus integration of the accessory genes. Currently, we do not have observations of enough evolutionary trajectories across different bacterial populations to draw these types of conclusions about conditional probabilities. New **Fig. 5** that we added to the discussion does more clearly show what types of cells can give rise to other types of cells, so that the evolutionary “flows” possible between them is shown more clearly.

5. In the fitness experiments, the deletion plasmid seems to be equally advantageous than chromosomal integration. However, the integration seems to be present in the population B6 during the experiment. Thus, the deletion plasmid may be favored over chromosomal integration. The authors should explain that (in the results or discussion). Same line of thoughts: Do the authors think the deletion plasmid might occur in the other populations at a later stage?

Reply: Our fitness assays tested strains in which we reconstructed integrations, deletion, and satellite plasmids in the ancestral *E. coli* cells. We did this to avoid complications that are present

in the evolution experiment: notably, most cells are likely to have secondary beneficial mutations in their chromosomes—in addition to the changes in the plasmid—by the end of the experiment, given that it had quite a long duration (~330 *E. coli* generations). That is, the eventual success of the deletion versus integration plasmids in population B6 cannot be predicted from the measured fitness effects of the plasmid mutations alone, because cells of each type have accumulated additional beneficial mutations. As described under **Comment #4**, this second wave of adaptation also occurs stochastically and will favor one or the other type of plasmid fate by chance as long as those cells are not too far behind in the evolutionary race. In population B6, the deletion appears to have gotten lucky by adding a very beneficial secondary mutation and swept to high frequency in this second round of evolution. Dynamics like this are normal in evolution experiments with large asexual bacterial populations.

We added an explanation of the presence and influence of secondary beneficial mutations by the end of the evolution experiment on Lines 323-325:

However, due to the long duration of the evolution experiment (~330 generations), other beneficial mutations are likely to have accumulated in the chromosomes of cells that have evolved plasmid sequences and influenced competition between each type by this time (42).

Therefore, to directly test the fitness consequences of evolving the different plasmid fates, ...

It is possible that deletion plasmids evolved and exist in population B2 at a low frequency that we cannot detect and that they could similarly get lucky and hitchhike to high frequency with a mutation that is very beneficial for growth, though the chances that they are “hiding out” below the detection threshold and have not been completely driven extinct decreases over time.

And why is the deletion observable in the *gfp* data but not in the sequencing reads for B2 (Fig. 2 C and E)?

Reply: The “Del” peaks in the flow cytometry data in **Fig. 2c** account for only a very small fraction of the total cells in these samples. The whole-genome sequencing data has a read-depth coverage of roughly 20-30× for the *E. coli* chromosome and 400-500× for the plasmid at its ancestral copy number. With this depth of sequencing, we are unlikely to detect the reads that span the new junction created in deletion plasmids if these plasmids are as rare as they appear to be in these samples from the flow cytometry data. Because we did not detect deletion plasmids in this way, we left a deletion line off of the graph in **Fig. 2e**. Note, however, that the PCR assay (which is biased toward detecting plasmids that produce shorter amplicons) does show faint bands on some days that have sizes consistent with deletion plasmids. It is also possible that cells that have lost GFP expression due to plasmid loss from segregation or due to recombination events in the chromosome that delete integrated plasmids contribute to the non-fluorescent events in this peak.

6. If I get it right, the objectives of the honey-bee gut experiments are to demonstrate the occurrence of SPs also in other systems (i.e., in ‘nature’). At the moment, this comes however quite as a surprise in the manuscript. The authors might want to elaborate on: what is the

justification to introduce the plasmid-carrying bacteria into the bee? Is it a specific environment where IncQ plasmids are known to occur?

Reply: You are correct. Given that one of the defining characteristics of IncQ plasmids is that they are broad-host-range plasmids, we wanted to test whether satellite plasmids evolved in a second bacterial host species to extend the generality of our findings. Honey bees have been proposed as a model system for studying bacterial-host interactions and their gut microbiome has beneficial effects on their health (Zheng et al. 2018; <https://doi.org/10.1038/s41684-018-0173-x>). To our knowledge, IncQ plasmids have not ever been reported in the honey bee gut microbiota. Rather, we turned to this system because we have prior experience using RSF1010 plasmids to genetically engineer many of these species, including *Gilliamella apicola*, *Snodgrassella alvi*, and *Bartonella apis* (Leonard et al. 2018; <https://doi.org/10.1021/acssynbio.7b00399>). As mentioned in the discussion, these experiments also let us determine whether we should be concerned about satellite plasmid evolution possibly complicating our engineering efforts when we use RSF1010 plasmids in these species in other ongoing projects (the answer: yes).

We added to and reworked the introduction to the bee section as follows on Lines 357-467:

Given the wide host-range of IncQ plasmids, we were interested in whether satellite plasmids would still evolve in a different bacterial species and in a more complex environment associated with an animal host. Honey bees (*Apis mellifera*) have a simple and conserved gut microbiome (42), and we previously genetically engineered several of its bacterial gut symbionts using RSF1010 plasmids (43). We tested whether SP evolution occurred in *Snodgrassella alvi* wkB2, a β -proteobacterium that has only recently been cultured (44). We first checked for the evolution of SPs from pQGS within five populations of *S. alvi* wkB2 (designated S1–S5) that we serially passaged *in vitro*.

7. The number of replicates tested $n=xx$ is sometimes missing and should be mentioned in the results or the figure legends (e.g. fitness experiments).

Reply: Thank you for pointing out this omission. We added the number of replicates to this legend and also now describe the type of error bars shown:

Error bars are 95% confidence intervals based on 15-18 replicate measurements for each strain.

We also added data points to **Fig. 2c** so that the statistics can be more readily evaluated by eye.

8. Sequencing results: a coverage plot of specific satellite plasmids would be a further proof for their existence. Can they be assembled from the sequencing reads?

Reply: We are not quite sure exactly which type of data/plot the reviewer would like to see. Perhaps these graphs of how many Illumina reads map to each position in the ancestral plasmid? The ones shown are for population B2 on Day 1 (left) and Day 4 (right). One can clearly see a loss of coverage corresponding to the deletion that is present in the satellite plasmid (pB253)

We did not purify satellite plasmid DNA away from the ancestral plasmid in the same cell and Illumina sequence it, so we could not de novo assemble it separately from the full-length plasmid DNA that is co-purified in our minipreps. We did Sanger sequence the PCR amplicons to determine the sizes of plasmids, and these are in agreement with restriction digests of plasmid samples (see the reply to **Comment #2** above and new **Fig. S1**). Their sizes are also in agreement with where we see changes in coverage graphs like the one shown above. Since this coverage information is conveyed by the summaries in **Fig. 2e–f** and the drawings of plasmids in **Fig. S3**, we do not feel that these types of coverage graphs add enough to the other information already presented in our manuscript to justify including them, but we are happy to add them to the supplement if the reviewer and editors think that they are necessary.

9. Overall, satellite plasmids seem to be transient while other solutions are favored over time. This is not in agreement with the title (and overall statements) as the accessory gene *gfp* is lost in some cases. The satellite plasmids may be selected because they reduce the fitness burden until the deletion/integration is fixed in the lineage.

Reply: This is correct. Satellite plasmids are transient evolutionary intermediates. Cells with satellite plasmids (Sat) will be outcompeted by cells with the plasmid integrated into the chromosome (Int) and/or cells with plasmids that delete the accessory genes while keeping the replication genes intact and therefore remaining autonomous (Del) as shown in **Fig. 2** and **Fig. 5**.

To better communicate this point, we changed the title to: “Evolution of satellite plasmids can **prolong** the maintenance of newly acquired accessory genes in bacteria”

We also changed the relevant statement in the abstract on Lines 25-28:

The evolution of satellite plasmids **appears to be transiently** favored relative to other plasmid fates that **eventually prevail** because **satellite plasmid evolution yields** a more immediate fitness advantage and because IncQ plasmids **may be particularly** prone to certain deletions during replication.

Reviewer #3 (Remarks to the Author):

In this manuscript, Zhang and colleagues investigate an intriguing evolutionary trajectory of a new plasmid-bacterial association. Using an elegant combination of experimental evolution, molecular biology, whole genome sequencing, genetic reconstruction, and stochastic modelling, the authors describe how the synthetic IncQ plasmid pQGS imposed a significant cost on the bacteria that host it, but these costs were driven down by the emergence of satellite plasmids that parasitise and compete with the original plasmid, suppressing copy number and thus cost. Overall I thought that the work was well-conducted and explained, with clear figures. Though satellite elements have been reported before, this study provides interesting context to their evolution, and thus will be of interest to a broad range of readers.

Main comments and suggestions:

1. My understanding is that in all populations, satellite plasmids emerged, but were transient. The ultimate fate of the plasmid was to either become integrated or to lose the accessory gene(s). The role of the satellite plasmids in stabilizing accessory genes therefore is essentially to provide an easily accessed, intermediate step on the fitness landscape that results in accessory genes being maintained longer than would be expected if the population immediately evolved the deletion plasmid. This idea — that plasmid evolutionary trajectory is determined partially by plasmid-plasmid competition allowing satellite plasmids to beat deletions in the short term — is really interesting. However, the transience of the satellite plasmids in host-plasmid evolution should be made clearer, at least in the abstract and in the summary. It currently reads as if satellite plasmids are relatively stable long-term (e.g. line 24-24 "the evolution of satellite plasmids relative to other plasmid fates") which is confusing.

Reply: We agree with these comments and made several edits to stress that satellite plasmids are transient and not long-term endpoints of plasmid evolution. These include changing “stabilize” to “prolong” in the title and editing the abstract. Please see the response to **Reviewer #2 Comment #9** who raised the same concern for the details of these changes.

A diagram showing the steps involved, the incremental fitness changes, the long-term outcomes, and the role of the satellite plasmid in this process, e.g. as a supplementary figure similar to Figure 7 in Cury et al. doi: 10.1093/molbev/msy123 or <https://natureecoevocommunity.nature.com/users/55681-michael-bottery/posts/18889-adapting-to-resistance> might also help.

Reply: This is a great suggestion. We created new **Fig. 5** for the discussion section to summarize the transient versus long-term fates of the transferred plasmids, their fitness consequences, how cells can potentially transition between them, and how this may impact the preservation of plasmids encoding accessory genes in nature. It is reproduced below.

Fig. 5 Evolutionary pathways resulting in the preservation or loss of accessory genes after plasmid transfer. This model summarizes various evolutionary fates of IncQ plasmids observed in this study and how they may relate to the maintenance of accessory genes on these plasmids in nature. After a new cell acquires an IncQ plasmid the evolution of nonautonomous satellite plasmids that are dependent on a full-length plasmid in the same cell for replication can alleviate the fitness burden of accessory genes encoded on the plasmid by reducing their copy number (Sat). These cells can still potentially benefit from the accessory genes if the environment changes (e.g., if they provide antibiotic resistance and there is treatment), and full-length plasmids in these cells remain capable of being transferred by conjugation to other cells to spread these accessory genes. Formation of satellite plasmids occurs at a high rate, but this state is likely to be a transient evolutionary intermediate. In experimental populations, cells with satellite plasmids are later displaced by cells that have integrated plasmid sequences into the chromosome via a mechanism involving an insertion sequence (Int) or the evolution of autonomous plasmids that delete just the accessory genes (Del). In the current study, cells had to maintain a gene from the plasmid that is encoded adjacent to the replication genes for survival (which is why no arrows are shown to the “no plasmid” state). Even without this constraint, the ability to evolve satellite plasmids is expected to give more opportunities for preserving the functions of accessory genes encoded on IncQ plasmids as they colonize new bacterial cells and hosts. The fitness and accessory gene copy number scales are not meant to be quantitative; they show only the approximate relative values of these parameters. Black arrows for transitions are roughly weighted by the relative rates inferred for these processes from the evolution experiments.

2. Again, the authors propose that satellite plasmid generation is a widespread mechanism in plasmid (particularly/exclusively IncQ plasmid) evolution for alleviating costs associated with acquisition of new accessory genes (lines 420-421). However, this seems to be in contrast to the statement that IncQ plasmids are evolutionarily stable (lines 89-90). I assume that this is because satellite plasmids are short-term/evolutionary dead ends: they are cheaper than ancestors, but are doubly vulnerable to loss by segregation as they are also lost if the co-resident ancestral plasmid they are parasitizing is lost? This could be made clearer.

Reply: Your logic is correct, but these are complementary rather than mutually exclusive statements. Cells with satellite plasmids arise at a high rate and are more-fit than their ancestors. This can allow the full-length plasmid to persist for longer in a newly colonized cell population, increasing the chances that it spreads the accessory genes to yet more cells/strains/species. Coupled with periodic exposure to environments that select for the accessory genes (the ones that are a useless burden in our experiments), the ability to form satellite plasmids could make full-length IncQ plasmids more widespread in nature than they would be without this mechanism that gives them a “quick release valve” that does not result in complete loss of the accessory genes.

The statement that IncQ plasmids are evolutionarily stable is based on observations in the literature that identical IncQ plasmids with multiple antibiotic resistance genes have been identified in different bacteria hosts and spanning a 30-year time interval, suggesting that IncQ plasmids are stable while transferring among different hosts in nature. Thus, it is referring to the overall stability of their lineage, not necessarily what happens after any given transfer. But, improving the ability to retain useful accessory genes contributes to keeping this lineage going by enabling the plasmid to carry useful cargo genes.

We added to the description of this process and cited **Fig. 5** in the discussion on Lines 468-473:

The improved fitness of cells with SPs makes it more likely that their offspring will survive and maintain copies of these accessory genes in the population until a future time when these genes are beneficial rather than burdensome. The ability to transiently evolve SPs creates a new pathway favoring preservation versus loss of accessory genes after a plasmid colonizes a new cell (**Fig. 5**). This evolutionary ‘safety valve’ can prolong the maintenance of accessory genes, such as those encoding antibiotic resistance, as a lineage of IncQ plasmids spreads and persists.

3. Is there any evidence of satellite plasmids in natural isolates? Could publicly available sequencing data be used to look for variance in coverage across IncQ plasmids?

Reply: This is a great idea. We also thought of this and looked into it before submitting this manuscript, but we did not have any success finding evidence for satellite plasmids in the very limited amount of IncQ plasmid next-generation DNA sequencing data that is currently available. Please see our first response to **Reviewer #1’s General Comments** for more details.

4. Figure S2. The authors might like to consider ordering the fragments by population rather than by size, so that the within- and between-population diversity can be assessed, as well as parallelism between clones from different populations.

Reply: We considered ordering the satellite plasmids in **Fig. S2** (now **Fig. S3**) by population. However, as only one or two SPs dominated in each population and we observed the same SPs in multiple populations, we prefer to keep them ordered by which parts of the plasmid they deleted.

5. It is not always clear whether PCR and sequencing is performed on individual clones or on the

whole population. E.g. line 171-172 I believe refers to the PCR assay conducted on the whole populations — this should be specified. In addition, the authors should specify how many clonal isolates were sequenced per population, either here or in the legend to Figure S2.

Reply: Thank you for pointing this out. We analyzed whole populations in the section in question. We have clarified whether PCR was performed on plasmids isolated from individual clones or whole populations in this section and also at later points, such as for the *S. alvi* experiments. We also added this information on the number of *E. coli* clonal isolates analyzed on Lines 220-222:

Two to five clones from each population were isolated and the PCR assay was used to determine whether they harbored SPs found in the population results. Sequencing the PCR amplicons from isolates that contained one of these new SP variants revealed...

6. Figure S2 is titled "Maps of evolved plasmids found in all experiments", which is confusing — I don't think these were found in all the experiments. In addition, I don't know how the authors can be sure they have exhaustively mapped all the evolved plasmids across all experiments. Perhaps "Maps of evolved plasmids identified in isolate clones" might be a better title?

Reply: Thank you for your correction. We have change the title of **Fig. S2** (now **Fig. S3** due to the addition of new **Fig. 1**) to simply “**Maps of evolved plasmids**”. Some of the plasmids found in *S. alvi* that are shown in the figure were sequenced from whole-population PCR products.

7. Figure 2 and Figure S3 show different times of sampling across the panels, which is confusing when trying to compare assays between timepoints. In addition, the x-axes are not always linear. This should be explained or at least mentioned in the legends.

Reply: We now point this out in these figure legends by adding the following statement:

The x-axes are categorical rather than linear in panels C-E, with even spacing of all samples that were analyzed from different days.

8. Line 195. It would help the reader to specify here that WGS was performed across the course of the experiment and to give the number of timepoints sequenced.

Reply: We added this information to the text as suggested on Lines 253-255:

Plasmid DNA isolated from cultures of the ancestral strain and whole-population samples archived at eight time points during the evolution experiment was examined by Illumina WGS.

9. The role of the IS sequence in integrating the plasmid is interesting and could bear a further sentence or two of discussion, e.g. how IS elements might facilitate capture/domestication of incompatible or otherwise unstable plasmids. See e.g. Lesic et al. doi: 10.1371/journal.pgen.1002529

Reply: We agree that this is an interesting observation that has also been seen in other experiments that examine plasmid evolution. We incorporated the new citation and call out this point more explicitly in discussion with this edited text on Lines 557-562:

IS elements have been broadly reported to facilitate the assimilation of genes carried on incompatible or unstable plasmids into bacterial chromosomes. Similar integration events have also been observed in experiments with different families of conjugative or suicide plasmids in *Pseudomonas putida* (10) and *Yersinia pseudotuberculosis* (62). In another instance, reversible chromosomal integration mediated by IS elements was reported for a virulence plasmid in *Shigella* (63).

10. Line 212-216 — could the hypothesis of multiple integrants be tested by qPCR on clones? At the moment this model is somewhat speculative.

11. Line 210 describes a PCR technique to test for integration. Ideally this should be extended to the other populations to support the flow cytometry data.

Reply to both comments: The observation of multicopy plasmid integrants is interesting and certainly stands out in the flow cytometry data. However, following up on these observations with additional PCR and qPCR experiments would take us quite far afield from the most novel point of our paper (characterizing satellite plasmids). We also do not think that it would add much knowledge beyond what we can already reasonably conclude from our existing data and what is known from other studies of plasmid evolution (see response to **Comment #9** above).

Those flow data clearly show that is common to see distinct categories of integrants from the peaks that have distinct “quantized” fluorescence levels (which we hypothesize result from one, two, three, etc., copies of the GFP gene). This is seen most clearly in populations in B3, B4, and B6 (**Fig. 2d**, **Fig. S4b**). Even though we can see bands in the PCR gels at these time points, the yield of plasmid DNA is very low from these populations when these integrant peaks dominate, further supporting that they result from integration events. We verify in one case in population B6 that *IS186* mediated integration is happening by PCR and the NGS results are consistent with this model. We provide several reasonable explanations for how multicopy integrants could arise, which we added to in the revised text here on Lines 275-284:

The presence of multiple peaks with intermediate fluorescence intensities—at the same time in population B6 (Fig. 2d) and later and to a lesser extent in population B2 (Fig. 2c)—indicates that the integrated plasmid sequence sometimes becomes multicopy within the chromosome. This could occur through homologous recombination during DNA replication, facilitated by the flanking *IS186* copies (Fig. 2g), leading to tandem duplications. Alternatively, plasmid sequences could integrate into multiple *IS186* copies or spread between them by gene conversion events.

We do already show some qPCR supporting the hypothesis of multicopy integrants, though this was not made clear in the original manuscript. To call attention to it, we also added this text to the same paragraph of the results on Lines 281-284:

qPCR of the original isolates (Fig. 1c) found one or three equal copies of *gfp* and *oriV* in clones B251 and B454 from day 5. Population B4 exhibits multiple and higher integrant fluorescence peaks at this time point (Fig. S4b), whereas population B2 does not (Fig. 2c), which is consistent with this model.

12. The authors expand the scope of their study to investigate plasmid evolution in a more natural system (honey bee gut), making their conclusions more general, which is great. However, all experiments are still performed with an artificial plasmid. Could satellite plasmid evolution be a feature of this artificial plasmid rather than of IncQ plasmids in general? A word or two on this would help readers understand the generality of the authors' findings.

Reply: Please see the reply to **Reviewer #2's general comments** for our full response to this question. In short, there is nothing especially unusual about the pQGS plasmid design. It is artificial inasmuch as it has reporter genes and our antibiotic resistance gene on it, but it has an organization similar to what one would expect for a natural IncQ/RSF1010 plasmid. We added a statement at the beginning of the results to point this out on Lines 144-145:

pQGS has three genes inserted into the RSF1010 backbone at a site where accessory genes are often found in natural IncQ plasmids (32).

13. Though they largely correlate, there are some inconsistencies between the different methods of assessing plasmid presence (PCR/FC/WGS) which should be mentioned/explained. For example, how do the authors explain the persistence of the band through to day 18 in 2A with the lack of 'satellite plasmid' fluorescence in 2C? The lack of a 'del' band in 2A given the low levels of del FC in 2C? Or the lack of 'satellite plasmid' fluorescence in 2D given the band in 2B?

Reply: We mentioned this, but clearly did not make the point strongly enough in the original manuscript: one should not directly compare the appearance of bands in the PCR assay (Fig. 2a–b) to the flow cytometry (Fig. 2c–d) or genome sequencing data (Fig. 2e–f). The PCR assay is biased toward smaller amplicons and cannot amplify all evolved plasmid fates. Because they make the smallest amplicons, one will get a satellite plasmid band even when SPs make up only a very small fraction of the population of plasmids in a sample—even when they are so rare that they are below the detection limit of the other methods. This is particularly true when cells with plasmid integrations dominate because there is no amplicon possible from these genomes (if it is in a single copy) or the amplicon will be too long to compete with the SP band (if the chromosome contains multiple tandem plasmid copies). PCR bias was a major reason that we used flow cytometry and whole-genome sequencing to more accurately track the dynamics.

PCR bias for SPs is related to the three discrepancies that the reviewer has noted:

1. Persistence of Sat band through Day 18 in population B2 even when Sat is not detected by flow cytometry past approximately Day 9 => This can be explained by the PCR being much more sensitive at detecting SP as explained above.

2. Lack of a Del band in population B2 given the low levels of events with no fluorescence seen in Fig. 2c. => There is a faint band here for deletion plasmid at some time points. In general, it will be overpowered by the shorter SP band in the PCR assay. Please see **Reviewer #2 Comment #5** for additional comments about the non-fluorescent population of cells in B2 and why they are not detectable by PCR and genome sequencing.
3. Lack of SP fluorescence in population B6 on Day 6 and later even when SP bands persist through day 15 => This can be explained by the PCR being much more sensitive at detecting SP as explained above.

We added text calling out the expected inconsistency between the PCR assay and other methods in several places in the text...

In the section of the results that introduces the flow cytometry and genome sequencing data (Lines 215-219):

SP formation was so widespread that multiple different SPs were even observed in the same population in two cases (B1 and B4). Note, however, that due to preferential PCR amplification of smaller DNA fragments, the relative intensities of different bands in these gels do not accurately reflect the representation of different molecular species in the sample. In particular, it can greatly overestimate the prevalence of SPs in a clone or population.

In the caption for **Fig. 2a–b**:

Compared to the methods shown in other panels, the PCR assay is biased toward detecting satellite plasmids versus other plasmid fates even when they are very rare in a population because they produce smaller amplicons.

14. Is it possible for satellite plasmids to parasitize deletion plasmids? Do the authors detect this at all?

Reply: Yes: satellite plasmids could evolve that parasitize the *gfp* deletion plasmid. They could evolve from the deletion plasmid simply by deleting one or more replication genes. However, we don't detect any evidence of this outcome in our experiments, and one might wonder why. We attribute this observation to the fact that evolution of the deletion plasmid has already alleviated most of the fitness burden of the plasmid on these cells. Therefore, the fitness benefit for evolving a deletion plasmid satellite is going to be small. There are likely to be much better beneficial mutations available to cells that enhance their growth in the nutrient environment of the evolution experiment at this point (e.g., by mutating some chromosomal gene to tune global gene regulation, as is commonly observed in bacterial evolution experiments). Cells with these better mutations would outcompete any that only evolved satellite deletion plasmids due to clonal interference in these strictly asexual *E. coli* populations. Eventually, satellite deletion plasmids could still evolve and predominate if their rate of formation is high enough that they can get "lucky" and evolve in a cell that has a cohort of other beneficial mutations.

15. The authors should consider including a few words about parasitic/satellite elements in other

systems in their discussion, e.g. in bacteriophage, see for example Frígols et al. 2015 doi: 10.1371/journal.pgen.1005609.

Reply: This is a great suggestion. Our decision to name the non-autonomous evolved plasmids “satellite plasmids” was motivated by knowledge of “satellite viruses”. We now explicitly make this connection in the discussion with this added text and cite the very interesting study by Frígols et al. that uses pathogenicity islands as models for satellite viruses on Lines 474-477:

Satellite viruses that require a helper virus to complete their lifecycles have been identified in microbes, plants, and animals (51, 52), and how bacterial viruses and their satellites co-evolve has been studied in laboratory evolution experiments (53). We report that bacterial plasmids can also evolve these types of molecular parasites.

Reviewers' Comments:

Reviewer #1:

Remarks to the Author:

The authors have successfully addressed my comments and I think the manuscript has improved thanks to the suggestions of the reviewers.

-line 99, accessory instead of "accessary"?

Reviewer #2:

Remarks to the Author:

The authors answered all comments and included some new analysis.

I have a couple of comments on the revised version that the authors/editor might want to consider prior to publication.

1. Regarding my previous comment 2:

The gel (Fig. S1) looks very interesting. It would be helpful to include the ancestral plasmid for comparison, as it would test if the satellite plasmids are already present at the onset of the experiment directly after introduction of the plasmid to the host (i.e., at a low frequency).

2. In retrospect – I would refrain the use of the term 'parasite' for the SPs – this because the authors did not go deeply into describing the bi-directional effect of the SPs on the large plasmid and the other way around. Notably, the SPs are probably more helping the persistence of the large plasmid (by reducing the dose effect) rather than parasitizing it.

3. The authors write now in the discussion:

'The improved fitness of cells with SPs makes it more likely that their offspring will survive and maintain copies of these accessory genes in the population until a future time when these genes are beneficial rather than burdensome.'

This statement is (in my taste) way to general to conclude from the presented experiments. All presented experiments were conducted in antibiotics – ie selection for the plasmid. Thus, the accessory gene that was required for survival was maintained on the deletion plasmid and the integration. In that sense – the general conclusion would be that the SPs emergence is another route for compensatory evolution in response to the fitness burden imposed by the plasmid (i.e., the accessory gene expression). The main message remains that there are two routes that ensure the stable maintenance of the resistance gene: losing the costly gfp or integration into the chromosome (lowering copy number).

Reviewer #3:

Remarks to the Author:

The authors have done a great job of responding to my and the other reviewers' comments; the manuscript is greatly improved and I would like to congratulate the authors on an interesting and well-written piece of work.

I have just two minor comments and I apologise for not raising these earlier.

1. Lines 17-18 suggests that compensatory mutations are "typically" associated with loss-of-function

in plasmid accessory genes but I do not think this is the case. Harrison et al. 2015 finds mutations in a global regulator; San Millan et al. 2015 and Loftie-Eaton et al. 2017 in chromosomal accessory helicases/kinases; Stalder et al. 2017 (and other Top lab papers on the IncP-1 pBP136) mainly in plasmid trfA; Porse et al. 2016 in plasmid conjugation machinery... Bottery et al. 2017 and Turner et al. 2014 do report inactivation of accessory genes, but I wouldn't consider this pattern 'typical'. I suggest that this sentence is toned down a bit, e.g. "Mutations that ameliorate this fitness cost can sometimes eventually stabilize a plasmid in a new host, but they can involve inactivating some of its novel accessory genes." (or similar)

2. Line 63 suggests that compensatory mutation takes a long time but a recent study shows it emerging more rapidly (<https://doi.org/10.1099/mic.0.000862>); a "but see (ref)" might be helpful here.

Reviewers' comments:

Reviewer #1 (Remarks to the Author):

The authors have successfully addressed my comments and I think the manuscript has improved thanks to the suggestions of the reviewers.

-line 99, accessory instead of “accessary”?

Reply: Thank you for pointing this out. We corrected the misspelling in the revised manuscript.

Reviewer #2 (Remarks to the Author):

The authors answered all comments and included some new analysis.

I have a couple of comments on the revised version that the authors/editor might want to consider prior to publication.

1. Regarding my previous comment 2:

The gel (Fig. S1) looks very interesting. It would be helpful to include the ancestral plasmid for comparison, as it would test if the satellite plasmids are already present at the onset of the experiment directly after introduction of the plasmid to the host (i.e., at a low frequency).

Reply: As suggested, we added a panel to **Supplementary Fig. 1b** showing plasmid DNA extracted directly from the ancestor strain. The only a band that is present corresponds to the full-length circular plasmid, so it is clear that no satellite plasmids are present at an appreciable frequency in the ancestral cell sample. Of course, a very small fraction of the cell population may have already evolved satellite plasmids by the time we grew this sample up from a single colony. From the gel, we roughly estimate that SPs are certainly present in $< 0.1\%$ of this population.

The updated **Supplementary Fig. S1** and its caption are reproduced on the following page.

Supplementary Figure 1. Direct visualization of satellite plasmids in evolved *E. coli* isolates. **a** Map of pQGS plasmid showing the single site cleaved by the restriction enzyme *ScaI* and the regions that are preserved in the satellite plasmids that are present in evolved strains B253, B353, and B653. **b** Total plasmid DNA isolated from the ancestor strain (Anc) and each of these evolved strains separated by gel electrophoresis. **c** Total plasmid DNA isolated from each of these evolved strains, digested with *ScaI*, and then separated by gel electrophoresis. After the circular pQGS ancestor plasmid present in these samples (Anc) is linearized by *ScaI*, the mobility of its band decreases, and it runs at the expected size of 9.3 kb relative to the DNA ladder. Similarly, the mobility of the satellite plasmid bands (Sat) decreases upon cleavage, which is consistent with the satellite plasmids being maintained as circular DNA molecules in these cells. After linearization, the satellite plasmid bands run at sizes that agree with the results of sequencing PCR products to determine which regions of the ancestral plasmid were deleted in each one. The nature of the molecules observed in the 1.0-1.5 kb size range relative to the DNA ladder is unclear. They are unaffected by *ScaI* treatment, and their apparent sizes track with the relative sizes of the satellite plasmids in each strain. They could be wholly or partially single-stranded DNA intermediates related to plasmid replication and/or transfer. Source data are provided as a Source Data file.

2. In retrospect – I would refrain the use of the term ‘parasite’ for the SPs – this because the authors did not go deeply into describing the bi-directional effect of the SPs on the large plasmid and the other way around. Notably, the SPs are probably more helping the persistence of the large plasmid (by reducing the dose effect) rather than parasitizing it.

Reply: We respectfully disagree with this comment. There is no doubt that SPs are molecular parasites of the ancestral plasmid. They depend on the presence of a full-length plasmid in the same cell for their continued replication because they exploit resources provided by the full-length plasmid (replication proteins). SPs reduce the full-length plasmid’s fitness when measured at the *level of its copy number within a cell*. The reviewer is correct that at the *level of a bacterial population*, the formation of SP parasites can improve the persistence and survival of the full-length plasmid. This is one of the main points of our paper. There is nothing in the definition of a ‘parasite’ that means that it cannot be beneficial *when considering other levels of selection or ecological interactions*. For example, there is evidence that parasitic intestinal worms modulate the immune system in ways that can actually benefit their human hosts—when one considers how the immune system interacts with other pathogens and the environment.

Furthermore, satellite viruses are often described in the literature as molecular parasites of their helper viruses in precisely the same way that we use the term here for plasmids. Thus, we feel that removing the term ‘parasite’ will actually make it more difficult for a reader to understand the place that satellite plasmids occupy in the DNA ecosystem that evolves within these cells.

3. The authors write now in the discussion:

‘The improved fitness of cells with SPs makes it more likely that their offspring will survive and maintain copies of these accessory genes in the population until a future time when these genes are beneficial rather than burdensome.’

This statement is (in my taste) way to general to conclude from the presented experiments. All presented experiments were conducted in antibiotics – ie selection for the plasmid. Thus, the accessory gene that was required for survival was maintained on the deletion plasmid and the integration. In that sense – the general conclusion would be that the SPs emergence is another route for compensatory evolution in response to the fitness burden imposed by the plasmid (i.e., the accessory gene expression). The main message remains that there are two routes that ensure the stable maintenance of the resistance gene: losing the costly *gfp* or integration into the chromosome (lowering copy number).

Reply: We believe that there is a small misunderstanding in how the reviewer is interpreting our experiments. Since this is least partially due to our failure to explain them as clearly as possible, we have taken this comment as an opportunity to further improve the manuscript.

In the discussion statement in question, we are discussing the fates of the burdensome ‘accessory genes’ that are not under selection. These are *lacI* and *gfp* in the model plasmid. This statement does not apply to the *aadA* (SpecR) antibiotic resistance gene, which is an essential, rather than an accessory gene under the conditions of the evolution experiment.

We updated the first paragraph of the Results to make this distinction clearer. We also labelled the *lacI* and *gfp* genes as “accessory genes” on **Fig. 1a**.

The constitutively expressed *gfp* gene and the adjacent *lacI* gene are not required for survival of these cells under the experimental conditions. They act as proxies for burdensome accessory genes. In a natural IncQ plasmid such genes might encode resistance to an antibiotic or other stressor that is not yet present in the environment but could be in the future. pQGS also encodes the *aadA* spectinomycin resistance gene at this site. Because we added spectinomycin to continuously select for maintenance of its function during the evolution experiment, *aadA* is an essential gene. It enables cells that acquired the plasmid to survive immediately, in their current environment, and cells without it will die.

If we are now on the same page after these clarifications, we don't understand how the reviewer can disagree with our conclusion that: ‘The improved fitness of cells with SPs makes it more likely that their offspring will survive and maintain copies [of plasmids] with these accessory genes in the population until a future time when these genes are beneficial rather than burdensome.’ (Note, we added “of plasmids” to this statement in the discussion to clarify the point further.) This conclusion follows logically from the state space and dynamics of the system rather than from any particular experiment. If one has a system in which satellite plasmids can form at an appreciable rate, this will lead to a sub-population of cells that maintain (fewer) copies of the accessory genes at less of a fitness cost. These cells will fare better in competition with cells that entirely lose the accessory genes or cells that lose the plasmids encoding the accessory genes (the Del and Int events describe in the results). SPs won't win in the long run against these fates in a population— unless the environment changes and cells with the accessory genes survive and those without them die — or unless the plasmid must remain transmissible and eventually move the accessory genes into a new host for that lineage of plasmids to survive.

Reviewer #3 (Remarks to the Author):

The authors have done a great job of responding to my and the other reviewers' comments; the manuscript is greatly improved and I would like to congratulate the authors on an interesting and well-written piece of work.

I have just two minor comments and I apologize for not raising these earlier.

1. Lines 17-18 suggests that compensatory mutations are "typically" associated with loss-of-function in plasmid accessory genes but I do not think this is the case. Harrison et al. 2015 finds mutations in a global regulator; San Millan et al. 2015 and Loftie-Eaton et al. 2017 in chromosomal accessory helicases/kinases; Stalder et al. 2017 (and other Top lab papers on the IncP-1 pBP136) mainly in plasmid *trfA*; Porse et al. 2016 in plasmid conjugation machinery... Bottery et al. 2017 and Turner et al. 2014 do report inactivation of accessory genes, but I wouldn't consider this pattern 'typical'. I suggest that this sentence is toned down a bit, e.g. "Mutations that ameliorate this fitness cost can sometimes eventually stabilize a plasmid in a new host, but they can involve inactivating some of its novel accessory genes." (or similar)

Reply: Thank you for suggesting this correction. We had to entirely remove this statement from the abstract when we shortened it to achieve the word limit.

2. Line 63 suggests that compensatory mutation takes a long time but a recent study shows it emerging more rapidly (<https://doi.org/10.1099/mic.0.000862>); a "but see (ref)" might be helpful here.

Reply: We added a citation to this very recently published study to this statement:

However, with some notable exceptions^{16,19}, it usually takes a long time—at least several hundred cell generations—and constant selection for plasmid function for compensatory mutations to arise when these mechanisms have been directly observed in laboratory populations of microbes²⁰.

The relevant references are:

16. Harrison, E., Guymer, D., Spiers, A. J., Paterson, S. & Brockhurst, M. A. Parallel compensatory evolution stabilizes plasmids across the parasitism-mutualism continuum. *Curr. Biol.* 25, 2034–2039 (2015).
19. Hall, J. P. J., Wright, R. C. T., Guymer, D., Harrison, E. & Brockhurst, M. A. Extremely fast amelioration of plasmid fitness costs by multiple functionally diverse pathways. *Microbiology* (2019). doi:10.1099/mic.0.000862
20. Harrison, E. & Brockhurst, M. A. Plasmid-mediated horizontal gene transfer is a coevolutionary process. *Trends Microbiol.* 20, 262–267 (2012).